# FRANCA: NESTED MATRYOSHKA CLUSTERING FOR SCALABLE VISUAL REPRESENTATION LEARNING

## ABSTRACT

We present $Franca$ (pronounced $Fran$-$ka$): *'free' one*; the first fully open-source (data, code, weights) vision foundation model that matches—and in many cases surpasses—the performance of state-of-the-art proprietary models, e.g., DINOv2, CLIP, SigLIPv2, etc. Our approach is grounded in a transparent training pipeline inspired from Web-SSL and uses publicly available data: Imagenet-21K and LAION-COCO (600M images). Beyond model release, we tackle critical limitations in self-supervised learning clustering methods. While recent models rely on assigning image features to large codebooks via clustering algorithms like Sinkhorn-Knopp, they fail to account for the inherent ambiguity in clustering semantics. To address this, we introduce a parameter-efficient, multi-head clustering projector based on nested Matryoshka representations. This design progressively refines features into increasingly fine-grained clusters without increasing the model size, enabling both performance and memory efficiency. Additionally, we propose a novel positional disentanglement strategy that explicitly removes positional biases from dense representations. This leads to consistent gains on several downstream tasks, demonstrating the utility of cleaner feature spaces.

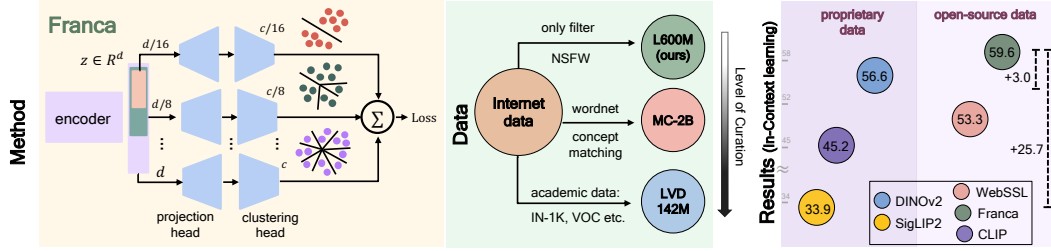

Figure 1: *Overview of* $Franca$. Left: We learn efficient *Matryoshka-style* (Kusupati et al., 2022) visual representations using a multi-head clustering projection head. The encoder produces features $z \in \mathbb{R}^d$, which is sliced into progressively smaller subsets of dimensions $d, \ldots d/8, d/16$. Each slice passes through a projection head and a corresponding clustering head with cluster counts $c, \ldots, c/8, c/16$, inducing a *coarse-to-fine hierarchy* of semantic abstraction. Middle: Unlike prior methods trained on proprietary data like WebLI in SigLIP 2 or curated academic datasets, e.g., LVD-142M in DINOv2, $Franca$ is trained on open-source internet-scale *minimally*-curated data. Right: It generalizes to dense tasks outperforming models trained on proprietary data.

## 1 INTRODUCTION

Self-supervised learning (SSL) offers a scalable approach to training Vision Foundation Models (VFMs) by leveraging the abundance of image-only data, which far exceeds the availability of paired image-caption data. This enables the learning of highly generalizable visual representations.

Despite the growing importance of VFMs, there is still a lack of *fully open, high-performing, and practical frameworks*. Current state-of-the-art models, including DINOv2 (Oquab et al., 2024), SEER (Goyal et al., 2021), billion-scale MAE (Singh et al., 2023), and SigLIP 2 (Tschannen et al., 2025), rely on proprietary datasets and often withhold critical or all training code, creating a significant barrier to reproducibility, accessibility, and scientific progress. To address this gap, we introduce $Franca$, a *fully open (data, weights, code) self-supervised VFM* that not only matches but often surpasses the performance of these proprietary counterparts. A key aspect of our contribution is the release of intermediate checkpoints, which provide unique insight into the training

| ATTRIBUTE / MODEL | METACLIP | WEB-SSL | SIGLIP 2 | AIMV2 | CLIP | DINOV2 | SAM | OPENCLIP | FRANCA |
|---|---|---|---|---|---|---|---|---|---|
| **Training Code/ Checkpoints** | | | | | | | | | |
| Model Publicly Available? | ✓ | ✓ | ✓ | ✓ | ✓ | ✓ | ✓ | ✓ | ✓ |
| Training Code Public? | ✓ | ✓ | ✗ | ✗ | ✗ | ✓ | ✗ | ✓ | ✓ |
| Intermediate Weights Public? | ✗ | ✗ | ✗ | ✗ | ✗ | ✗ | ✗ | ✓ | ✓ |
| **Training Data** | | | | | | | | | |
| Training Data Public? | ✓ | ✓ | ✗ | ✗ | ✗ | ✗ | ∼ | ✓ | ✓ |
| Training Data Downloadable? | ✗ | ✗ | ✗ | ✗ | ✗ | ✗ | ∼ | ✓ | ✓ |
| Data Deduplication Code Public? | ✗ | ✗ | ✗ | ✗ | ✗ | ✗ | ✓ | ✓ | ✓ |
| Data NSFW & CSAM Filtered? | ✗ | ✗ | ✓ | ? | ? | ? | ? | ✓ | ✓ |

Table 1: *Openness of Visual Foundation Models.* We analyze various models based on the public availability of their components MetaCLIP (Xu et al., 2024), Web-SSL (Fan et al., 2025), SigLIP 2 (Tschannen et al., 2025), AIMv2 (Fini et al., 2025), CLIP (Radford et al., 2021), DINOv2 (Oquab et al., 2024), SAM (Kirillov et al., 2023) and OpenCLIP (Cherti et al., 2023; Nezhurina et al., 2025). Franca exemplifies a fully *open-source* approach, providing complete transparency from model weights to data and processing methods. ∼: partially; ?: not specified. NSFW and CSAM are acronyms for "Not Safe For Work" and "Child Sexual Abuse Material," respectively.

trajectory by enabling the community to analyze convergence behavior and study emergent properties. Inspired by Web-SSL's openness, Franca provides a more accessible framework for models of different scales, intermediate checkpoints and data, while also achieving superior performance as detailed in our experiments. *By integrating full openness, high performance, and practical accessibility, Franca establishes a new standard for transparent VFM research* as shown in Table 1.

The strong performance of Franca stems from two key technical innovations that overcome fundamental limitations in SSL. The first addresses a core shortcoming of models such as DINOv2, which depend on optimal-transport clustering (i.e., Sinkhorn-Knopp) for pseudo-label assignment. This process is inherently ambiguous—for instance, vehicles can be grouped by manufacturer, color, or model year—and current methods address this by using very large codebooks (e.g., 131K in DINOv2), which may not generalize well across domains. To address this, we introduce a multi-head clustering projector using nested Matryoshka representations (Kusupati et al., 2022), where progressive neuron subsets cluster data into increasingly finer-grained groupings. This approach not only reduces parameters compared to conventional approaches but also improves performance and decreases memory requirements for downstream tasks such as k-nearest neighbors classification, leading to higher performances at equal memory.

Second, we address a subtle issue with dense clustering: representations can be biased by patch position rather than semantic content. We introduce a lightweight post-pretraining technique that first learns linear projections to predict patch positions, then projects the latent space to an orthogonal subspace devoid of this positional information. The result is a dense representation space that emphasizes semantics over spatial positioning, leading to substantial gains on challenging benchmarks like dense in-context learning (Balazevic et al., 2023a) and unsupervised semantic segmentation (Van Gansbeke et al., 2021). In summary, our key contributions are:

- We present Franca, the first fully open-source (code, data, weights) and high-performance VFM that often outperforms proprietary models while ensuring clear accessibility

- We introduce two technical innovations: a Matryoshka multi-head clustering approach for efficient, high-quality representations, and a spatial-semantic disentanglement post-pretraining technique that refines representations for stronger backbones. Together, these methods significantly advance the base SSL approach, as shown in our analysis (see Figure 2b).

- We demonstrate strong performance across diverse tasks: including in-context learning, linear segmentation and object discovery, surpassing DINOv2-G by up to 3%, outperforming it on OOD detection and 3D understanding, while matching its performance on classification—all without proprietary data.

## 2 METHOD

We propose Franca, a scalable open-source self-supervised learning (SSL) framework built on DINOv2 (Oquab et al., 2024) and pretrained on large public image datasets. It tackles key limitations in existing vision SSL models through three main components. First, *CyclicMask*, inspired

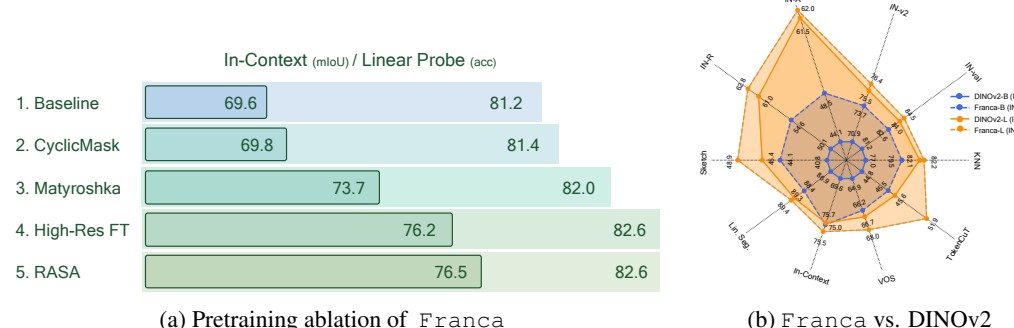

(a) Pretraining ablation of `Franca`    (b) `Franca` vs. DINOv2

Figure 2: *Ablation study and comparison with DINOv2 with ImageNet-21k pretraining. (a) Ablation:* We incrementally add CyclicMask, Matryoshka representations, RASA, and high-resolution fine-tuning to a DINOv2-B pretrained on ImageNet-21k. Each addition yields consistent performance gains, measured by linear probing on ImageNet-1K (outer bar) and in-context segmentation (Balazevic et al., 2023a) on Pascal VOC (inner bar). *(b) Controlled Comparison:* Under an equal training setup on IN-21K (no high-resolution fine-tuning or distillation), `Franca` significantly outperforms DINOv2 across ViT-B and ViT-L backbones on a suite of tasks encompassing classification, robustness, and dense prediction.

by (Darcet et al., 2024), is a masking strategy that circularly shifts masked patches to break simple spatial continuity and promote semantic feature learning. Second, we introduce *Matryoshka Embeddings* (Kusupati et al., 2022), a nested multi-head clustering approach that shares projection layers to generate compressed multi-resolution representations; and finally *RASA*, a lightweight post-pretraining step that removes feature components correlated with absolute patch positions, resulting in spatially invariant representations. Figure 2a shows that each component provides consistent gains in both in-context segmentation and linear classification. These cumulative gains enable `Franca` to outperform DINOv2 across a suite of tasks under an equal IN-21K training setup (see Figure 2b).

**Preliminaries** We adopt the multi-crop training strategy from DINO (Caron et al., 2021). An input image is transformed into multiple augmented views (global and local crops). Each view $x$ is then split into $n$ non-overlapping patches, which are embedded into $\mathbb{R}^d$, and a classification token ([CLS] $\in \mathbb{R}^d$) is prepended to form the input sequence. A Vision Transformer (ViT) backbone (Dosovitskiy et al., 2021) processes this sequence, producing $n + 1$ embeddings ($n$ patch embeddings and one [CLS] embedding). The same ViT architecture is shared between the student $f_\theta$ and teacher $\bar{f}_{\bar{\theta}}$, producing $Z_s = f_\theta(x) \in \mathbb{R}^{(n+1) \times d}$, $Z_t = \bar{f}_{\bar{\theta}}(x) \in \mathbb{R}^{(n+1) \times d}$, where $Z_s$ represents the student's output embeddings and $Z_t$ represents the teacher's embeddings. The teacher's parameters $\bar{\theta}$ are updated via exponential moving average (EMA) of the student's parameters.

For supervision, we apply projection heads to the student embeddings $Z_s$. The [CLS] embedding is passed through a DINO-style head (a 3-layer MLP with softmax over prototypes) that produces image-level prototype scores, while the patch embeddings are processed by an iBOT-style head that produces patch-level prototype scores. For brevity, we denote both heads as $h_\theta$ for the student and $\bar{h}_{\bar{\theta}}$ for the teacher (same architecture, EMA-updated). The teacher's projected outputs are clustered using Sinkhorn-Knopp (Cuturi, 2013) to produce balanced target distributions. The student is trained to match these targets via cross-entropy loss, denoted as $\mathcal{L}$.

## 2.1 MATRYOSHKA REPRESENTATIONS FOR EFFICIENT MULTI-GRANULAR LEARNING

Standard self-supervised models produce fixed-size embeddings, limiting flexibility under different compute or downstream constraints. To enable adaptable representations across feature granularities, we adopt Matryoshka representations (Kusupati et al., 2022), which nest progressively truncated subspaces of a high-dimensional embedding.

Formally, let $Z_s = f_\theta(x) \in \mathbb{R}^{(n+1) \times d}$ be the unmodified ViT's output (patch + [CLS] embeddings). We define nested dimensions $\mathcal{M} = \{m_1, \ldots, m_k\}$, where $m_1 < \cdots < m_k = d$, and extract sub-embeddings $Z_s^{(j)} = Z_s[:, 1 : m_j], \quad \forall m_j \in \mathcal{M}$.

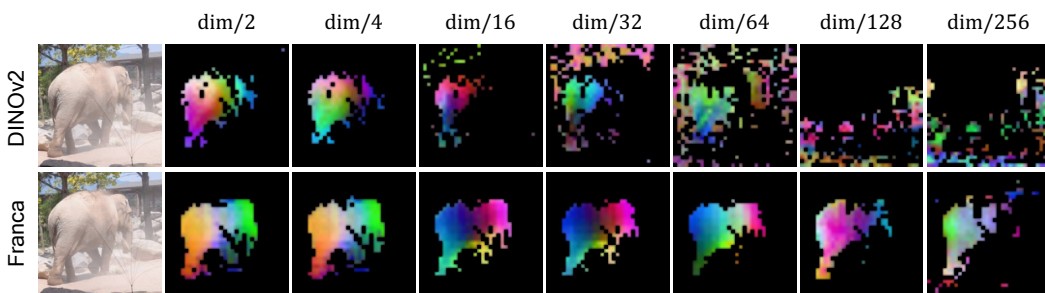

Figure 3: *PCA visualizations across Matryoshka slices.* First three PCA components for different feature slices $m_j$ of Franca and DINOv2. Despite Franca being trained only up to dim/16, it maintains coherent part structure even in smaller feature dimension as compared to DINOv2.

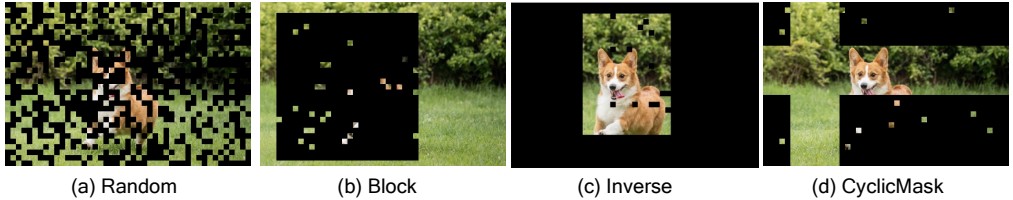

Figure 4: *Masking strategies used in masked image modeling.* Compared to Random (a), Block (b), and Inverse (c) masking, our CyclicMask (d) circularly shifts the visible region across spatial axes, preventing the model from being biased toward specific spatial locations.

Each $Z_s^{(j)}$ is processed by an independent projection head $h_v^{(j)}$, with proportionally fewer prototypes as $m_j$ decreases. A cross-entropy loss $\mathcal{L}^{(j)}$ is applied to each head's output. The total loss is the sum across all levels with equal weights: $\mathcal{L}_{\text{total}} = \sum_{j=1}^{k} \mathcal{L}^{(j)}$. The standard Matryoshka approach (Kusupati et al., 2022) slices the whole *encoder*'s output along the feature dimension and applies the *same* projection head to each sub-embedding. In contrast, we keep the backbone unmodified and extend this setup by attaching a *dedicated projection head and clustering objective* to each subspace. This allows each slice to produce distinct prototypes and prototype assignments, encouraging specialization across representational granularity of the features across training steps. Our framework supports hierarchical learning: coarse heads capture global semantics, while fine heads focus on local structure akin to early clustering works (Ji et al., 2019; Asano et al., 2020; Van Gansbeke et al., 2020) and unlike most recent representation learning works that optimize only a single feature space (Oquab et al., 2024; Tschannen et al., 2025; Radford et al., 2021).

As shown in Figure 2a, our Matryoshka framework yields the largest gains on dense prediction tasks and the PCA visualizations in Figure 3 show that Franca preserves coherent part-level structure beyond trained dimensions, whereas DINOv2 loses semantic alignment.

## 2.2 BALANCING SPATIAL DISTRIBUTION OF VISIBLE PATCHES WITH CYCLICMASK

Masked image modeling trains models to reconstruct masked patches, typically using random or block masking (Oquab et al., 2024; Zhou et al., 2022a;b). These strategies often produce fragmented visible regions with limited contextual coherence, as shown in Figure 4 (a-b). Inverse block masking (Baevski et al., 2023) improves continuity by retaining a central visible block Figure 4 (c), but introduces positional bias since central patches are always observed. We propose CyclicMask, inspired by (Darcet et al., 2024), which applies random cyclic shifts to the inverse block mask along both spatial axes. The pseudo-code for this augmentation is given in Algorithm 1. This simple modification preserves contiguous context while ensuring uniform exposure across all patch positions. As shown in Figure 2a, CyclicMask improves baseline performance on IN-1K linear probing and in-context segmentation (Pariza et al., 2024) by 0.2%.

## 2.3 RASA: REMOVAL OF ABSOLUTE SPATIAL ATTRIBUTES

ViT models often develop unintended spatial biases from their fixed patch layouts and positional embeddings, entangling location with semantic content. Our preliminary study in Figure 5 demon-

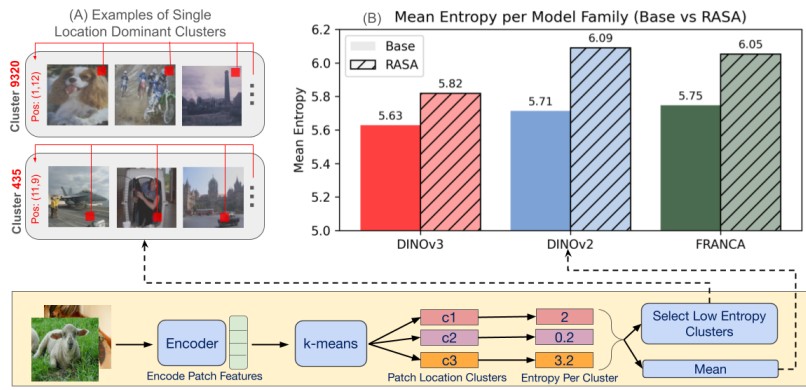

Figure 5: *Entropy of patch locations per cluster.* For each visual cluster out of the $65k$ clusters from $k$-means, we compute the entropy of the 2D coordinates of its assigned patches (bottom row). Lower Mean Entropy (B: top right) indicates positional bias in its representations. We find that DINOv2 has many low-entropy clusters firing at fixed locations (A: top left). Top right: Our RASA post-training increases spatial entropy, thereby debiasing positional information from patch features.

strates this issue: using a frozen model on COCO images, we assigned each visual patch to $65k$ clusters from $k$-means and computed the spatial entropy of patch locations per cluster (Figure 5: bottom). The results show that patch clusters are frequently centered at fixed positions (Figure 5-A: top-left), exhibiting low mean spatial entropy (Figure 5-B: top-right). These indicate that cluster assignment is often driven more by location than by semantics. Our subsequent RASA post-training enhances the spatial entropy (Figure 5-B: top-right), forming less location-driven clusters.

To directly address the spatial entanglement, we propose Removal of Absolute Spatial Attributes (*RASA*), a post-training method designed to disentangle spatial information from patch embeddings. After pretraining, we process the patch features $Z$ (for brevity, we denote $Z_t$ as $Z$ throughout this section) through an alternating optimization procedure. At iteration $t$, the input consists of the $n$ patch embeddings $Z^{(t)} = \{Z_i \in \mathbb{R}^D\}_{i=1}^n$. We first optimize a linear position prediction head, parametrized by a matrix $W \in \mathbb{R}^{2 \times D}$, on a small set of images to predict normalized patch coordinates via a sigmoid $\sigma(\cdot)$, minimizing the mean squared error:

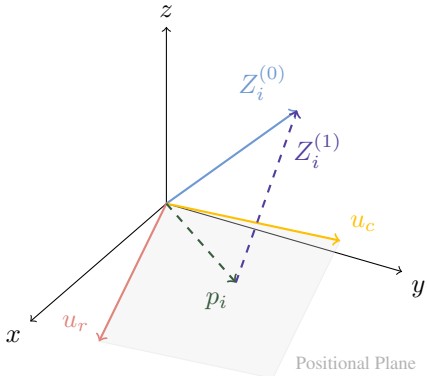

Figure 6: Each iteration of RASA projects a patch embedding $Z_i$ onto a learned positional plane $\text{span}\{u_r, u_c\}$ and subtracts its projection $p_i$.

$$\mathcal{L}_{\text{pos}} = \frac{1}{n} \sum_{i=1}^{n} \|\sigma(W Z_i) - y_i\|_2^2, \quad (1)$$

where $y_i \in [0, 1]^2$ are the normalized 2D coordinates of patch $i$. The row vectors of $W$ are orthonormalized using Gram–Schmidt (Golub & Van Loan, 2013) to form the basis vectors $u_r$ and $u_c$, which span the positional subspace (visualized in gray in Figure 6). We then remove the component of each feature vector that lies within this positional subspace. This is achieved by projecting $Z_i$ onto the subspace and subtracting that projection, yielding a refined embedding $Z_i^{(t+1)}$ that is less aligned with positional information while retaining semantic content:

$$p_i = \langle Z_i, u_r \rangle u_r + \langle Z_i, u_c \rangle u_c, \quad (2)$$

$$Z_i^{(t+1)} = Z_i^{(t)} - p_i^{(t)}. \quad (3)$$

This iterative refinement process, summarized in Figure 6, continues until $\mathcal{L}_{\text{pos}}$ saturates (typically within 9 iterations), effectively removing linearly predictable spatial bias while preserving semantic

| METHOD | BACKBONE | LIN. SEG. | | IN-CONTEXT | |
|---|---|---|---|---|---|
| | | VOC | ADE20K | VOC | ADE20K |
| SigLIP | ViT-B/16 | 57.8 | 23.1 | 33.9 | 10.6 |
| SigLIP 2 | ViT-B/16 | 69.6 | 23.1 | 65.0 | 32.3 |
| iBOT | ViT-B/16 | 73.1 | 30.1 | 66.6 | 26.9 |
| EVA-CLIP | ViT-B/16 | 70.4 | 34.6 | 34.8 | 11.3 |
| DINOv2[†] | ViT-B/14 | 86.9 | 41.3 | 69.6 | 30.0 |
| DINOv2[§] | ViT-B/14 | **90.2** | **49.7** | **77.1** | **37.7** |
| Franca (ours) | ViT-B/14 | 89.4 | 46.2 | 76.5 | 35.0 |
| SigLIP 2 | ViT-L/16 | 66.6 | 29.8 | 46.4 | 21.3 |
| Web-SSL | ViT-L/14 | **92.3** | 46.3 | 71.3 | 35.3 |
| DINOv2[†] | ViT-L/14 | 89.3 | 45.4 | 72.0 | 33.5 |
| DINOv2[§] | ViT-L/14 | 90.3 | **50.7** | 74.6 | 38.6 |
| Franca (ours) | ViT-L/14 | 90.5 | 48.9 | **79.5** | **39.6** |
| Web-SSL | ViT-G/14 | 89.5 | **46.7** | 73.3 | 36.7 |
| DINOv2 | ViT-G/14 | **90.6** | 46.2 | 73.7 | **37.7** |
| Franca (ours) | ViT-G/14 | 90.2 | 46.5 | **76.7** | 36.5 |

(a) Linear and In-context Segmentation.

| METHOD | ARCH. | VIDEO OBJ. SEGM. $(\mathcal{J} + \mathcal{F})_m$ | TOKENCUT | |
|---|---|---|---|---|
| | | | VOC07 | VOC12 |
| RADIOv2.5 | ViT-B/16 | 66.7 | 44.6 | 48.1 |
| C-RADIOv3 | ViT-B/16 | 67.7 | 47.3 | 50.8 |
| DINOv2[§] | ViT-B/14 | 68.5 | 42.9 | 48.3 |
| DINOv2-R[§] | ViT-B/14 | 67.8 | 53.0 | 56.9 |
| Franca | ViT-B/14 | **70.6** | **53.2** | **59.1** |
| RADIOv2.5 | ViT-L/16 | 67.5 | 51.3 | 54.6 |
| C-RADIOv3 | ViT-L/16 | 67.3 | – | – |
| DINOv2[§] | ViT-L/14 | **69.1** | 41.3 | 47.2 |
| DINOv2-R[§] | ViT-L/14 | 66.9 | 57.6 | **61.3** |
| Franca | ViT-L/14 | **69.1** | 59.5 | 61.0 |

(b) Unsupervised Video Object Segmentation on DAVIS and Object Discovery using TokenCuT.

Table 2: *Segmentation Benchmarks.* Left: Linear and In-Context Segmentation. Right: Video object segmentation on DAVIS2017 and Unsupervised Object Discovery using TokenCuT (VOC 07/12). [†]: reproduced on IN-21K, without distillation; [§]: distilled from DINOv2(-R)-G on LVD-142M. **bold**: best; underline: second best

content. Due to its linear construction, the entire RASA transformation for a single iteration $t$ can be expressed as multiplication by a single matrix $L^{(t)}$:

$$Z_i^{(t+1)} = Z_i^{(t)} L^{(t)} = Z_i^{(t)} (I - p_i^{(t)}) = Z_i^{(t)} (I - u_r u_r^\top + u_c u_c^\top), \qquad (4)$$

Furthermore, the complete multi-step transformation is simply the product of these matrices, $L = \prod_{t=1}^T L^{(t)}$. This final matrix $L$ can be absorbed into the weights of the final ViT layer, resulting in no architectural changes and zero inference overhead.

## 3 EXPERIMENTAL RESULTS

**Training setup** We train Franca using ViT-B, ViT-L, and ViT-G encoders with patch size 14 and without registers. We pretrain from scratch for 625K iterations with batch size 2048 for ViT-B and 3072 for ViT-L and ViT-G using an image resolution $224 \times 224$ with *Matryoshka Embeddings* and *CyclicMask*. ViT-L and ViT-G is pretrained on LAION-600M, while ViT-B (for ablations) uses ImageNet-21K. We then finetune the models with batch size 1024 at $364 \times 364$ and $518 \times 518$ resolutions (for 30K and 10K iterations, respectively) on a mix of ImageNet-1K, ADE20K, COCO, KITTI, and VOC; due to computational constraints, ViT-G skips this stage. Finally, we apply the lightweight *RASA* post-training on Pascal VOC, for 8 iterations, using a learning rate of 0.002, and a batch size of 128. More details are in Appendix C.

### 3.1 CONTROLLED COMPARISON WITH DINOv2 ON IMAGENET-21K

We begin by conducting a controlled comparison between Franca and DINOv2 to study the effect of architectural differences. We use ViT-B/14 and ViT-L/14 backbones pretrained on IN-21K with identical hyperparameters—and crucially, without high-resolution fine-tuning or distillation from a ViT-G model—we isolate the impact of Franca's architectural and training innovations. As shown in Figure 2b, Franca consistently outperforms DINOv2 across a suite of tasks, with the largest gains observed on robustness benchmarks (IN-A, IN-R, Sketch) and dense segmentation tasks (ADE20K, TokenCut), demonstrating that the Matryoshka, CyclicMask, and RASA components lead to more semantically aligned ViT representations that are more appropriate for dense tasks.

### 3.2 DENSE TASKS

**In-Context Learning** We evaluate Franca on the Hummingbird Benchmark (Balazevic et al., 2023a; Pariza et al., 2024), which frames semantic segmentation as a nearest-neighbor retrieval task using patch features, without model fine-tuning. We report mean Intersection-over-Union (mIoU) on Pascal VOC (Everingham et al.) and ADE20K (Zhou et al., 2017) in Table 2a.

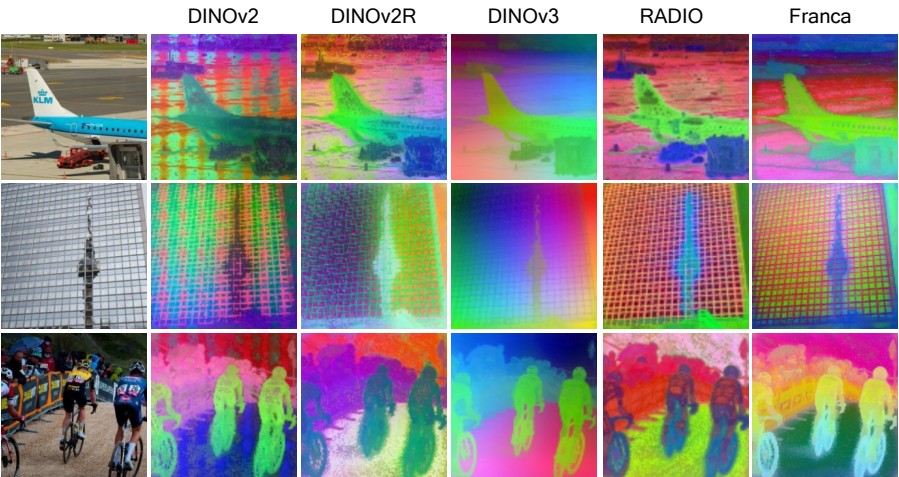

Figure 7: *Semantic Consistency of Dense Features.* Visualization of dense features via PCA (first three components as RGB) demonstrates that our `Franca` features consistently match semantic parts with the same color, despite variations in pose, style, or even object identity (e.g., the consistent coloring for the bikers and bikes in the last row). Images were randomly sampled from the Segment Anything-1B dataset (Kirillov et al., 2023) (`np.random.randint(seed=42)`).

`Franca` consistently achieves strong segmentation results. For instance, with ViT-L/14, it surpasses DINOv2-L (distilled from DINOv2-G and trained on LVD-142M) and Web-SSL by up to +5% and +8% mIoU on Pascal VOC, and by +1% and +4% on ADE20K, respectively. Notably, DINOv2's pretraining data (LVD-142M) includes VOC and ADE20K, whereas `Franca` only uses them during the limited high-resolution finetuning stage (which `Franca`'s ViT-G skips). Thus, `Franca` is exposed to the training splits of these evaluation datasets far less than DINOv2 and yet produces superior results. Also, the performance gains of `Franca` increase with model capacity. These results demonstrate `Franca`'s ability to learn spatially precise, semantically meaningful features that transfer effectively to segmentation without fine-tuning. We also report overclustering performance, which measures semantic alignment of spatial features in Table 8.

**Linear segmentation** Table 2a also evaluates `Franca` on semantic segmentation under a linear probing protocol. `Franca` achieves strong results across all backbones. With ViT-B/14, it performs on par with DINOv2-B, which is distilled from DINOv2-G and pretrained on LVD-142M. Crucially, a non-distilled DINOv2-B model trained on ImageNet-21K (the same setting as `Franca` for ViT-B) performs substantially worse (86.9 vs. 89.4 mIoU on VOC; 41.3 vs. 46.2 on ADE20K). This suggests the strong performance of the official DINOv2-B is largely due to distillation. With ViT-G/14, `Franca` matches DINOv2 and outperforms Web-SSL. Notably, `Franca` surpasses Web-SSL on several benchmarks despite Web-SSL's pretraining on a substantially larger dataset (2B images), highlighting our approach's effectiveness. Interestingly, SigLIP 2 (Tschannen et al., 2025) performs poorly on segmentation, particularly on VOC (57.8 mIoU), suggesting limited spatial localization capabilities despite its extended training and strong image-text alignment.

**Video Object Segmentation** We extend our evaluation to video object segmentation (VOS) on DAVIS (Pont-Tuset et al., 2017), where the goal is to propagate a ground-truth mask from the first frame through a video using feature similarity. From Table 2b, `Franca` achieves the best results across all backbones. With ViT-B, it scores 70.6%, surpassing DINOv2 by 2%. With ViT-L, `Franca` matches or outperforms other methods, reaching 69.1%. This shows that `Franca` produces temporally stable features that enable consistent tracking.

**Unsupervised Object Discovery** We further evaluate unsupervised object discovery using Token-Cut (Wang et al., 2022b) in Table 2b, which segments objects by leveraging patch-level feature similarity. On VOC07, `Franca`-B achieves a higher CorLoc than DINOv2. On VOC12, `Franca` obtains 59.1% CorLoc, outperforming DINOv2 (48.3%) and DINOv2-R (56.9%). This demonstrates that `Franca` provides spatially coherent features that facilitate unsupervised object segmentation.

| METHOD | ARCH. | DATA | TEXT | KNN | IN-VAL | v2 | IN-A | IN-R | Sketch |
|---|---|---|---|---|---|---|---|---|---|
| | | | | | CLASSIFICATION | | | ROBUSTNESS | |
| IBoT | ViT-B/16 | IN-21K | ✗ | 77.1 | 79.5 | – | – | – | – |
| DINOv2[†] | ViT-B/14 | IN-21K | ✗ | 77.0 | 81.2 | 70.9 | 44.1 | 50.1 | 40.8 |
| DINOv2[§] | ViT-B/14 | LVD-142M | ✗ | **82.1** | **84.5** | **75.1** | **55.1** | **63.3** | **50.6** |
| Franca (ours) | ViT-B/14 | IN-21K | ✗ | 79.5 | 82.6 | 73.7 | 48.5 | 54.6 | 44.1 |
| SigLIP | ViT-L/16 | WebLI | ✓ | – | 80.5 | 74.2 | 76.5 | 95.0 | 73.6 |
| SigLIP 2 | ViT-L/16 | WebLI | ✓ | – | 82.5 | 76.8 | 84.3 | **95.7** | **75.5** |
| PE$_{core}$ | ViT-L/16 | MC-2B | ✓ | **83.5** | 83.9 | 77.9 | **89.0** | 95.2 | 73.4 |
| Web-SSL | ViT-L/16 | MC-2B | ✗ | 76.8 | 82.4 | 71.0 | 67.3 | 68.9 | 54.8 |
| DINOv2[†] | ViT-L/14 | IN-21K | ✗ | 82.1 | 84.0 | 75.5 | 61.5 | 61.0 | 45.4 |
| DINOv2[§] | ViT-L/14 | LVD-142M | ✗ | **83.5** | **86.3** | **78.0** | 71.3 | 74.4 | 59.3 |
| Franca (ours) | ViT-L/14 | LAION-600M | ✗ | 81.6 | 84.9 | 76.0 | 66.5 | 64.3 | 56.6 |
| OpenCLIP | ViT-G/14 | LAION-2B | ✓ | **83.2** | **86.2** | 77.2 | 63.8 | **87.8** | **66.4** |
| DINOv2 | ViT-G/14 | LVD-142M | ✗ | 83.1 | 86.0 | **77.9** | **75.9** | 78.8 | 62.5 |
| Web-SSL | ViT-G/14 | MC-2B | ✗ | 79.2 | 84.7 | 74.3 | 73.3 | 75.9 | 60.9 |
| Franca (ours) | ViT-G/14 | LAION-600M | ✗ | 83.0 | 86.0 | **77.9** | 75.6 | 75.8 | 60.6 |

Table 3: *Classification and Robustness.* performance across vision-language and vision-only models. We report top-1 linear probing accuracy on IN-1K (val, ReaL, v2) and robustness benchmarks (IN-A, IN-R, Sketch). Franca, trained without text supervision, matches or exceeds the performance of larger text-supervised models and outperforms DINOv2 when reproduced on the same data and training strategy. †: reproduced on IN-21K without distillation; §: distilled from DINOv2-G on LVD-142M. **Bold**: Best; Underline: second best.

**PCA of Patch Features** We visualize patch tokens by projecting them into a 3D RGB color space using PCA. As shown in Figure 7, the resulting color maps reveal a stark contrast. For DINOv2, PCA highlights only a few scattered high-variance patches (outliers), failing to cover the full object and yielding a fragmented, noisy segmentation. In contrast, Franca produces dense, coherent color segments aligned with the actual object. Its contours are sharply delineated, and semantically similar parts (e.g., bike, body of aeroplane) receive consistent colors across instances—patterns that do not emerge in DINOv2. Importantly, neither model was trained on the Segment Anything-1B (Kirillov et al., 2023) subset used for these visualizations, confirming that the segmentation patterns are entirely emergent from the self-supervised features. We also visualize the self-attention of different models in Figure 9, showing that Franca's attention maps provide better object localization.

## 3.3 CLASSIFICATION AND ROBUSTNESS

**Image Classification** We evaluate the global image representations learned by Franca for image classification using nearest-neighbor (KNN) and linear probing protocols on ImageNet-1K. For linear probing, we report results on both the standard validation set and ImageNet-v2 (Recht et al., 2019). As shown in Table 3, Franca achieves performance comparable to DINOv2, even for ViT-B and ViT-L despite the fact that for these variants Franca does not use distillation from a ViT-G teacher (as DINOv2 does), and the ViT-B model is trained only on the 13M images of IN-21K. Franca also outperforms Web-SSL (Fan et al., 2025), which uses a much larger dataset (MetaCLIP-2B). Notably, our Franca-G model matches the linear accuracy of Web-SSL-7B (85.9% vs. 86.0%) with 7× fewer parameters.

To ensure a fair comparison, Table 3 and Figure 2b include DINOv2 models (ViT-B/14, ViT-L/14) re-trained on IN-21K without distillation from a ViT-G teacher, using our training setup. These models perform significantly worse than the original DINOv2, indicating that the ViT-G distillation contributes substantially to their performance. In contrast, Franca achieves strong results without such distillation or extra supervision, matching or surpassing these reproduced DINOv2 versions.

**Robustness** We assess robustness by applying the linear classification heads to the validation sets of ImageNet-A (Hendrycks et al., 2021b), ImageNet-R (Hendrycks et al., 2021a), and Sketch (Wang et al., 2019), which introduce semantic or stylistic variations. From Table 3, Franca demonstrates strong robustness, where Franca-G matches DINOv2-G across all three datasets and outperforms OpenCLIP-G by 9% on ImageNet-A, despite OpenCLIP being trained on over 30× more data.

**Out-of-Distribution Detection** We further evaluate Franca on out-of-distribution (OOD) detection using the OpenOOD benchmark (Zhang et al., 2024) reporting the Area under the ROC curve (AuROC) metric across five datasets. Results in Figure 8 show that Franca consistently outper-

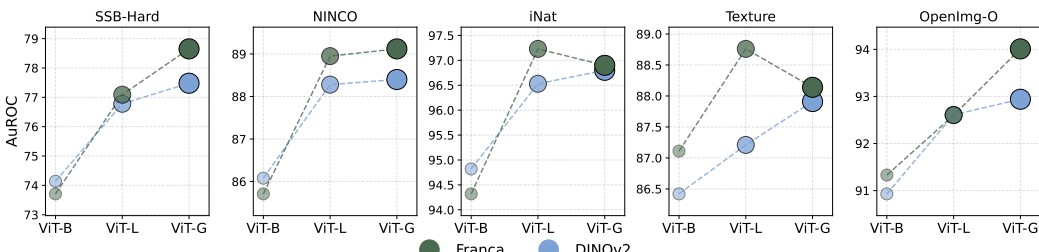

Figure 8: *Out-of-Distribution Detection* across five robustness-benchmarks: SSB-Hard (Vaze et al., 2022), NINCO (Bitterwolf et al., 2023), iNaturalist (Huang & Li, 2021), OpenImage-O (Wang et al., 2022a), and Texture (Kylberg, 2011). Franca consistently outperforms DINOv2, at larger scales, demonstrating its robustness across distribution shifts. DINOv2-B and DINOv2-L are distilled from DINOv2-G and trained on LVD 142M, while neither variants of Franca are distilled.

| MODEL | GEOMETRY | | | TEXTURE | | | GEOMETRY+TEXTURE | | |
|---|---|---|---|---|---|---|---|---|---|
| | PSNR ↑ | SSIM ↑ | LPIPS ↓ | PSNR ↑ | SSIM ↑ | LPIPS ↓ | PSNR ↑ | SSIM ↑ | LPIPS ↓ |
| EVA-CLIP | 19.01 | 0.62 | 0.35 | 18.02 | 0.60 | 0.32 | 18.85 | 0.63 | 0.38 |
| CLIP | 19.47 | 0.64 | 0.33 | 18.04 | 0.60 | 0.32 | 19.42 | 0.64 | 0.39 |
| SigLIP 2 | 19.39 | 0.64 | 0.33 | **18.11** | 0.60 | **0.31** | 19.36 | 0.64 | 0.41 |
| AIMv2 | 19.16 | 0.62 | 0.35 | 18.05 | 0.60 | 0.32 | 19.04 | 0.63 | 0.39 |
| WebSSL | 19.56 | 0.64 | 0.33 | 18.05 | 0.60 | 0.32 | 19.43 | 0.64 | 0.37 |
| DINOv2 | 19.34 | 0.63 | 0.34 | 17.96 | 0.60 | 0.33 | 19.31 | 0.64 | **0.36** |
| Franca | **19.58** | **0.65** | **0.32** | 17.97 | **0.62** | 0.32 | **19.53** | **0.65** | 0.37 |

Table 4: *Probing with Gaussian Splatting*, Normalized average metrics using Feat2GS (Chen et al., 2025) across six datasets for two probing schemes: Geometry, Texture and All: Geometry + Texture with ViT-L backbone. We measure PSNR, SSIM (higher is better) and LPIPS (lower is better). Franca outperforms SoTA VFMs suggesting strong geometrical awareness.

forms DINOv2 across large and giant model variants, demonstrating strong robustness to distribution shifts and effective scaling for OOD detection.

## 3.4 PROBING 3D AWARENESS

We evaluate texture and geometry awareness using the Feat2GS framework (Chen et al., 2025), which leverages novel view synthesis as a proxy for 3D understanding. Features from visual foundation models are mapped into 3D Gaussians through a lightweight readout trained with a photometric loss. For fair comparison, inputs are resized to 512, feature maps are upsampled to 512, and PCA reduces dimensionality to 256. Evaluation uses PSNR, SSIM, and LPIPS under three setups: (a) Geometry—features predict geometric parameters while texture is optimized; (b) Texture—features predict texture while geometry is optimized; and (c) All—features predict both geometry and texture. We compare Franca with DINOv2, WebSSL, AIMv2, EVA-CLIP, SigLIP 2, and CLIP across six datasets, averaging results over five runs. From Table 4, Franca achieves the best performance in Geometry and All, indicating strong 3D geometric awareness, while all methods perform similarly on Texture, with SigLIP 2 slightly ahead. We also show the results on dense keypoint matching and depth prediction in Table 7.

## 4 CONCLUSION

In this work, we present Franca, a new Vision Foundation Model that is open-weight, open-data and open-code. We build this model using a novel Matryoshka-nested clustering self-supervised loss that allows for learning hierarchical representations and have introduced RASA, a simple post-pretraining method to remove overtly spatial biases in the final representations. Across evaluations in image recognition, segmentation, robustness, OOD detection and 3D understanding, we find that it matches and frequently outperforms DINOv2 and other state-of-the-art models such as SigLIP 2. The recent release of DINOv3 (Siméoni et al., 2025) is a concurrent development. While it represents a significant advance, initial analysis (Figure 5, Figure 7) suggests that positional bias remains a challenge, indicating the continued relevance of our technical contributions like RASA. Integrating our innovations into frameworks like DINOv3 is a promising direction for future work.

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

CONTENTS

## A  APPENDIX

## B  RELATED WORK

Our work builds on and contributes to four major areas of prior research: self-supervised learning for visual representation learning, scaling strategies for data and model capacity in vision models, open-source foundation models, and techniques for disentangling semantic content from positional or representational biases.

**Self-Supervised Learning (SSL) for Vision.**   Self-supervised learning has emerged as a powerful paradigm for visual representation learning without any manual annotations. By designing pretext tasks that utilize image structure as supervision signals, SSL methods enable models to learn transferable features. Early approaches used handcrafted objectives such as context prediction (Doersch et al., 2015), patch reordering (Noroozi & Favaro, 2016), colorization (Zhang et al., 2016; 2017), inpainting (Pathak et al., 2016), geometric transformation prediction (Gidaris & Komodakis, 2018), and instance discrimination (Dosovitskiy et al., 2014; Wu et al., 2018). Modern SSL methods primarily focus on learning invariances across augmented data views. While early approaches leverage contrastive learning (Oord et al., 2018; Misra & Maaten, 2020; Chen et al., 2020a; He et al., 2020; Chen et al., 2020b; 2021) by aligning positive pairs and separating negatives, bootstrap-based (Grill et al., 2020; Chen & He, 2021; Gidaris et al., 2021) and distillation-based methods (Caron et al., 2021; Oquab et al., 2024) refine targets through teacher-student networks, often removing the need for negative pairs. More recently, Masked Image Modeling (MIM) has emerged as a dominant SSL strategy, where models learn to reconstruct masked patches (He et al., 2022; Zhou et al., 2022a; Kakogeorgiou et al., 2022; Bao et al., 2022; Wei et al., 2022). Beyond these, clustering-based methods (Caron et al., 2018; 2020; Ji et al., 2019; Sirko-Galouchenko et al., 2025) have gained prominence, assigning pseudo-labels through algorithms like K-means or Sinkhorn-Knopp. The combination of MIM with clustering has shown particular promise, as exemplified by recent works

such as MOCA (Gidaris et al., 2024) and CAPI (Darcet et al., 2025). This hybrid approach leverages the strengths of both paradigms.

While current vision foundation models often fall into categories like vision-language or MAE-like architectures, which have their own strengths and limitations (e.g., reliance on text supervision or need for task-specific adaptations), DINOv2 (Oquab et al., 2024) stands out as a powerful pretrained model employing a clustering-based approach. However, DINOv2 has two key limitations: as a clustering method, it doesn't inherently capture the ambiguity often present in assignments at a fixed granularity, nor does it explicitly incorporate the benefits of modern hierarchical masking strategies. Our work addresses these concerns by integrating nested *Matryoshka* projections (Kusupati et al., 2022) directly into its objective. This allows each subspace to perform clustering at a *different* granularity, yielding diverse pseudo-labels efficiently (see Figure 1). Combined with an improved input masking strategy, our approach enables the joint learning of coarse-to-fine semantics without increasing model size, leading to strongly improved performances and reduction in memory.

**Open Foundation Vision Models.** The reliance on proprietary datasets in the training of current vision foundation models raises critical concerns regarding transparency, reproducibility, and the disentanglement of contributions. Models such as SEER (Goyal et al., 2019), DINOv2 (Oquab et al., 2024), CLIP (Radford et al., 2021), and billion-scale MAE (Singh et al., 2023) are all trained on proprietary data. This practice makes it challenging for the research community to isolate the true impact of model's novelty and training strategies from the unique characteristics and biases of the datasets themselves. The lack of access to these datasets hinders independent verification, fair comparison, and a comprehensive understanding of what truly drives model performance. Inspired by the success and large-scale utilization of CLIP, OpenCLIP (Ilharco et al., 2021; Cherti et al., 2023) and MetaCLIP (Xu et al., 2024; Chuang et al., 2025) are the first to release models trained on public data. OpenCLIP (Cherti et al., 2023; Nezhurina et al., 2025) stands out with the fully reproducible pipeline leveraging the ready-to-use LAION dataset (Schuhmann et al., 2022; LAION, 2024), study of scaling laws and release of intermediate checkpoints. On vision-only foundation models, the performance of DINOv2 has proven difficult to match by public data models. Web-SSL (Fan et al., 2025) extends the study of large-scale self-supervised pretraining by training models on publicly available MetaCLIP-2B (Xu et al., 2024) dataset, showing that models trained on open data can approach the performance of those trained on proprietary data on VLM tasks, but still below DINOv2 on visual perception tasks.

Building on this, we present a fully open-source vision foundation model using publicly available datasets, ReLAION (LAION, 2024), as it represents the most popular and safe public dataset for large-scale vision model training.

**Spatial correlations in learned representations.** A common issue in dense self-supervised learning is the entanglement of semantic content with positional cues, causing models to rely on location rather than object identity. For instance, a model trained on "cows in grassy fields" and "camels in deserts" may misclassify a cow on a beach as a camel, due to learned associations with background context (Arjovsky et al., 2019). Such spatial biases reduce generalization and can hinder performance when objects appear in atypical locations (e.g., a cow in the sky) (Singh et al., 2020).

Several works have addressed this by proposing methods invariant to positional information. Lenc & Vedaldi (2015) enforce equivariance to geometric transformations; Wang et al. (2023) disentangle representations into orthogonal subspaces for content and style. Invariant Risk Minimization (Arjovsky et al., 2019) seeks features stable across environments, minimizing reliance on spurious cues. We propose a simple post-training strategy that learns a linear projection to identify and remove spatial information from features. Since, we use it as a post-training strategy, it requires no architectural changes and can be easily adapted to any pretrained model to reduce spatial bias.

## C  IMPLEMENTATION DETAILS

A summary of the different datasets used at different training stage in given in Table 5

**Pretraining Datasets** We pretrain Franca on large-scale, publicly available image-only datasets to ensure full reproducibility. We use the ImageNet-21K (Ridnik et al., 2021), which contains approximately 13.1M high-quality, hierarchically labeled images across 21,841 classes. This dataset

| Training type | Architecture | Dataset | #Images |
|---|---|---|---|
| Pretraining | ViT-B | ImageNet-21K | 13,153,000 |
| Pretraining | ViT-L | LAION-COCO | 599,187,600 |
| Pretraining | ViT-G | LAION-COCO | 599,187,600 |
| High-Res Finetuning | ViT-B, ViT-L | IN-1K, ADE20K, COCO, VOC, KITTI | 1,444,270 |
| RASA Post-training | ViT-B/L/G | Pascal VOC | 17,125 |

Table 5: Summary of training setup across architectures, datasets, and image counts.

---

**Algorithm 1** Pseudo-code for CYCLICMASK

---

**Require:** input size $(H, W)$, masking ratio $m \in [0, 1]$, roll flag `roll`, aspect ratio range $[r_{\min}, r_{\max}]$

1: $T \leftarrow H \times W$        ▷ total number of patches
2: $n \leftarrow \lfloor m \cdot T \rfloor$        ▷ number of masking patches
3: $c \leftarrow \max(1, T - n)$        ▷ number of complement patches
4: $r \sim \exp\left(\mathcal{U}[\log r_{\min}, \log r_{\max}]\right)$        ▷ sample aspect ratio
5: $h \leftarrow \min\left(H, \lceil \sqrt{c \cdot r} \rceil\right)$
6: $w \leftarrow \min\left(W, \lceil \sqrt{c/r} \rceil\right)$
7: `top` $\sim \mathcal{U}\{0, H - h\}$,   `left` $\sim \mathcal{U}\{0, W - w\}$
8: `mask` $\leftarrow$ `torch.zeros(`$H, W$`,, dtype=bool)`
9: `mask[top:top + h, left:left + w]` $\leftarrow$ `True`
10: `mask` $\leftarrow \neg$ `mask`        ▷ invert block to mask everything else
11: **if** `roll` = `True` **then**
12:      `shift_x` $\sim \mathcal{U}\{0, H - 1\}$,   `shift_y` $\sim \mathcal{U}\{0, W - 1\}$
13:      `mask` $\leftarrow$ `torch.roll(mask, (shift_x, shift_y), dims=(0,1))`
14: **end if**
15: **return** `mask` of size $H \times W$

---

offers broad visual coverage and is widely used in foundation model pretraining. To further scale up training and improve generalization, we also leverage LAION-600M[1], a subset of ReLAION-2B, which is a research-safe version of the LAION-5B dataset (Schuhmann et al., 2022; LAION, 2024). While LAION-5B is originally paired image-text data, we discard the text and use only the image modality.

**Training** `Franca`'s architecture follows DINOv2 (Oquab et al., 2024) without registers, using Vision Transformers (Dosovitskiy et al., 2021) of varying model capacities: ViT-B with 86M parameters, ViT-L with 300M, and ViT-G with 1.1B. All models are trained from scratch for 625K iterations without distillation from larger models, unlike DINOv2, which distills from ViT-G into smaller variants. We use CyclicMask (pseudo-code in Algorithm 1) and employ Matryoshka (Kusupati et al., 2022) with five nested heads with feature dimensions $[d, \frac{d}{2}, \ldots, \frac{d}{16}]$ on top of the normal ViT backbone.

For LAION-600M, we use global crops scale of $[0.48, 1.0]$, following DINOv2-style augmentations. Stochastic depth regularization is set to 0.1 for ViT-B and 0.4 for ViT-L and ViT-G. We use a total batch size of 2048 for ViT-B and 3072 for both ViT-L and ViT-G, distributed across 32, 64 and 128 H100 GPUs for ViT-B, ViT-L and ViT-G respectively. The learning rate is set to $1 \times 10^{-3}$ for the Base model and $3.5 \times 10^{-4}$ for the Large and Giant variants, using a cosine schedule with warmup of 100K iterations.

We train RASA on top of our frozen backbone on Pascal VOC using crops of resolution $518 \times 518$, batch size of 128, with AdamW (Loshchilov & Hutter, 2019) optimizer. For each of the 8 incremental head training iterations, we used a dual-head linear projection (one for predictings and the other the y-axis patch positions) with sigmoid activation. In every iteration, only the top head was trained for 5 epochs with no weight decay and an initial learning rate of $2 \times 10^{-3}$, while all heads from previous iterations are absorbed into the weights of the final ViT layer, for iteratively removing the positional information from the features of the last ViT layer.

---

[1] https://huggingface.co/datasets/laion/laion-coco

**High-resolution adaptation** We initialize the model with pretrained weights and perform incremental high-resolution finetuning. First, we finetune the model at input resolution $364 \times 364$ with a local crop resolution of $112 \times 112$, using a base learning rate of $1.25 \times 10^{-5}$ for 30K iterations. The resulting checkpoint is then used to initialize the model for a second finetuning stage at input resolution $518 \times 518$ with a local crop size of $168 \times 168$, again with the same learning rate, for 10K iterations. All schedules are preserved but temporally compressed to fit within the shorter training horizons. The teacher network undergoes a warmup phase during the first 10K iterations of the initial 364-resolution stage to stabilize early training dynamics. We use a dataset mix comprising only the training set of ImageNet-1K, ADE20K, COCO, KITTI, and VOC. Due to computational constraints, high-resolution finetuning is performed only for the ViT-B and ViT-L model.

## D  EVALUATION DETAILS

**Overclustering** In the overclustering setting, we follow the protocol of (Ziegler & Asano, 2022). Specifically, we perform $K$-Means clustering (via faiss (Johnson et al., 2019)) on the spatial tokens extracted from the backbone, discarding the projection head. To align clusters with ground-truth semantic labels, we first apply greedy matching based on pixel-level precision and then refine the assignment with Hungarian matching (Kuhn, 1955), ensuring permutation-invariant evaluation as in (Ji et al., 2019). Inputs are cropped to $448 \times 448$, while clustering operates on downsampled $100 \times 100$ masks to reduce the computational cost of Hungarian matching. Results are reported as the mean Intersection-over-Union (mIoU), averaged over five seeds, across twp datasets: COCO-Thing, and Pascal VOC 2012 (Everingham et al.).

**Visual In-Context Learning** The Dense Nearest Neighbor Retrieval Evaluation, introduced by (Balazevic et al., 2023b) and openly implemented by (Pariza et al., 2024), is designed to measure the scene understanding ability of dense image encoders through a retrieval-based protocol. The evaluation proceeds in three stages:

1. **Memory Bank Construction**: Given a training dataset with dense annotations, two memory banks are built. The first stores patch-level features obtained from the spatial outputs of a dense encoder, while the second stores the corresponding patch-level labels.

2. **Query Processing**: For each validation image, we extract patch embeddings from the encoder's spatial output. Each query patch searches for its $k$ nearest neighbors in the feature memory bank, and the associated labels of these neighbors are aggregated to infer the query patch label.

3. **Evaluation**: After generating a predicted dense annotation for the full image, the result is compared against ground-truth labels to compute performance.

Since the original implementation of (Balazevic et al., 2023b) is not publicly available, we rely on the open-source reimplementation from (Pariza et al., 2024), which follows the authors' description and allows leveraging either the ScaNN library (Guo et al., 2020) or the faiss library (Johnson et al., 2019) for efficient nearest-neighbor search. In our experiments, we adhere closely to this setup but make two modifications: (1) instead of restricting memory to a fixed capacity of 10,240,000 entries, we index all patch embeddings extracted from images resized to $518 \times 518$; and (2) we employ GPU-accelerated FAISS for nearest-neighbor retrieval, configured to approximate the ScaNN setup used in the original evaluation (e.g., with $k = 30$ neighbors).

We report results as mean Intersection-over-Union (mIoU) on two benchmark datasets: Pascal VOC 2012 (Everingham et al.) and ADE20K (Zhou et al., 2017), averaging over five random seeds.

**Linear Segmentation** Linear segmentation is a common protocol to assess the linear separability of learned spatial representations. The setup freezes the encoder and trains a lightweight segmentation head, typically a single linear layer with batch normalization, using pixel-wise cross-entropy loss. In practice, the spatial patch embeddings produced by the backbone are bilinearly upsampled to match the resolution of the ground-truth masks, and the linear head is trained on top. This procedure is implemented in `mmsegmentation` and was used in the DINOv2 paper (Oquab et al., 2023) for evaluating Pascal VOC and ADE20K, where the backbone is kept frozen and only the linear head is optimized.

In this paper, we adopt the DINOv2 setup as a baseline and introduce a few modifications tailored to our experimental needs:

- **General changes (both datasets):** We switch from iteration-based training (`max_iters = 40000`) to epoch-based training with a maximum of 50 epochs. The crop size is increased from $512 \times 512$ to $518 \times 518$, and the stride is adjusted accordingly to half the crop size. Data augmentation uses `RandomResize` instead of `Resize`. The learning rate schedule is also defined by epoch rather than iteration.

- **Pascal VOC 2012 (Everingham et al.):** We replace the AdamW optimizer (used in DINOv2) with SGD (lr = 0.001, momentum = 0.9, weight decay = $5 \times 10^{-4}$). The resize target is set to $(2048, 518)$ with a ratio range of $(0.5, 2.0)$, consistent with the crop size change.

- **ADE20K (Zhou et al., 2017):** We keep AdamW as the optimizer but wrap it with an `OptimWrapper`. The random resize ratio range is expanded from $(0.5, 2.0)$ to $(1.0, 3.0)$, making the scale augmentation more aggressive. At test time, instead of single-scale evaluation, we adopt multi-scale testing with image ratios $[1.0, 1.32, 1.73]$.

These modifications preserve the core linear segmentation evaluation protocol of DINOv2 while adapting it for our framework and experimental goals. We generally have seen the modifications to work better and improve further the results of all the models we tested.

**Evaluation Datasets**    **Pascal VOC 2012** (Everingham et al.). We use the most recent `trainaug` split, which contains 10,582 annotated images across 21 categories, including one background class. The validation set includes 1,449 images. We exclude unlabeled objects and the boundary class from evaluation. For hyperparameter tuning of the fully unsupervised segmentation method of (Ziegler & Asano, 2022), we rely on the original `train` split (1,464 images).

**COCO-Stuff 164K** (Caesar et al., 2018). The full COCO-Stuff dataset provides semantic annotations for both "stuff" (background, amorphous regions) and "thing" (foreground, countable objects) categories, with 91 stuff and 80 thing classes, respectively. It contains 118,000 training images and 5,000 validation images. In prior work (Ziegler & Asano, 2022; Pariza et al., 2025), two variants of this dataset are considered:

1. *COCO-Stuff*, where the 91 stuff categories are grouped into 15 broader classes (with all thing categories collapsed into a single "other" label that is ignored during evaluation).

2. *COCO-Thing*, where the 80 thing categories are consolidated into 12 broader object classes, and background regions are excluded from evaluation.

In this work, we only make use of the *COCO-Thing* variant. Specifically, we follow (Kirillov et al., 2019) to obtain panoptic instance annotations, which are then merged into object-level categories using the official conversion script. The resulting 12-class setup emphasizes object-level reasoning in cluttered natural scenes and serves as our benchmark for overclustering experiments. Although we refer to the dataset as "COCO-Stuff 164K" for consistency with the literature, only the COCO-Thing portion is used in our evaluations.

**ADE20K** (Zhou et al., 2017). ADE20K is a large-scale scene parsing benchmark with 150 semantic categories, ranging from background classes (e.g., sky, grass) to fine-grained objects (e.g., person, car). The dataset contains 20,210 training images and 2,000 validation images with detailed annotations. Due to its diversity and fine-grained structure, ADE20K is widely regarded as one of the most challenging datasets for dense prediction tasks. In our experiments, we use the full dataset but ignore the *others* label during evaluation.

# E  ADDITIONAL RESULTS

## E.1  FINE-GRAINED CLASSIFICATION

We evaluate the transferability of the learned representations on 11 classification benchmarks introduced in SimCLR (Chen et al., 2020a). These benchmarks cover a variety of tasks, including scene

| Method | Arch | Food | C10 | C100 | SUN | Cars | Aircr | DTD | Pets | Cal | Flwrs | CUB | Avg |
|---|---|---|---|---|---|---|---|---|---|---|---|---|---|
| MAE | ViT-H/14 | 78.4 | 96.1 | 83.9 | 63.9 | 56.1 | 63.4 | 75.4 | 89.4 | 95.9 | 92.3 | 57.2 | 77.5 |
| DINOv2[†] | ViT-L/14 | 93.4 | 99.2 | 93.9 | 78.1 | 89.9 | 81.7 | 82.9 | 95.2 | 87.2 | 99.6 | 90.3 | 90.1 |
| DINOv2[§] | ViT-L/14 | 94.3 | 99.3 | 93.4 | 78.7 | 89.9 | 81.5 | 84.0 | 96.5 | 97.5 | 99.7 | 90.5 | 91.4 |
| Web-SSL | ViT-L/14 | 91.0 | 98.9 | 90.7 | 77.5 | 88.9 | 80.2 | 83.6 | 93.1 | 95.1 | 98.8 | 90.9 | 89.9 |
| DINOv2 | ViT-G/14 | 94.7 | **99.5** | 94.4 | 78.7 | 91.4 | **87.2** | 84.5 | 96.7 | **97.6** | **99.7** | **91.6** | **92.3** |
| Web-SSL | ViT-G/14 | 94.1 | 99.4 | 93.1 | 78.0 | 90.3 | 83.7 | 84.7 | 92.4 | 96.8 | 99.4 | 91.2 | 91.2 |
| OpenCLIP | ViT-G/14 | 94.5 | 98.7 | 91.0 | 84.0 | **96.1** | 80.2 | **86.0** | 95.7 | 98.1 | 99.5 | 89.9 | 92.2 |
| Franca (ours) | ViT-B/14 | 90.6 | 98.7 | 90.9 | 77.0 | 88.7 | 75.2 | 81.7 | 94.1 | 96.2 | 99.7 | 86.2 | 88.9 |
| Franca (ours) | ViT-L/14 | 94.3 | 99.4 | 94.1 | 79.9 | 89.5 | 81.3 | 84.1 | 95.1 | 97.4 | 99.8 | 91.1 | 91.5 |
| Franca (ours) | ViT-G/14 | **95.0** | 99.5 | **95.1** | 78.9 | 91.3 | 85.5 | 85.0 | **97.2** | 97.5 | **99.7** | 91.3 | **92.3** |

Table 6: *Linear evaluation of frozen features on fine-grained datasets.* top-1 accuracy measured across 11 benchmarks across objects, scenes, and textures, following (Chen et al., 2020a); [†]: reproduced on IN-21K without distillation; [§]: distilled from DINOv2-G on LVD-142M.

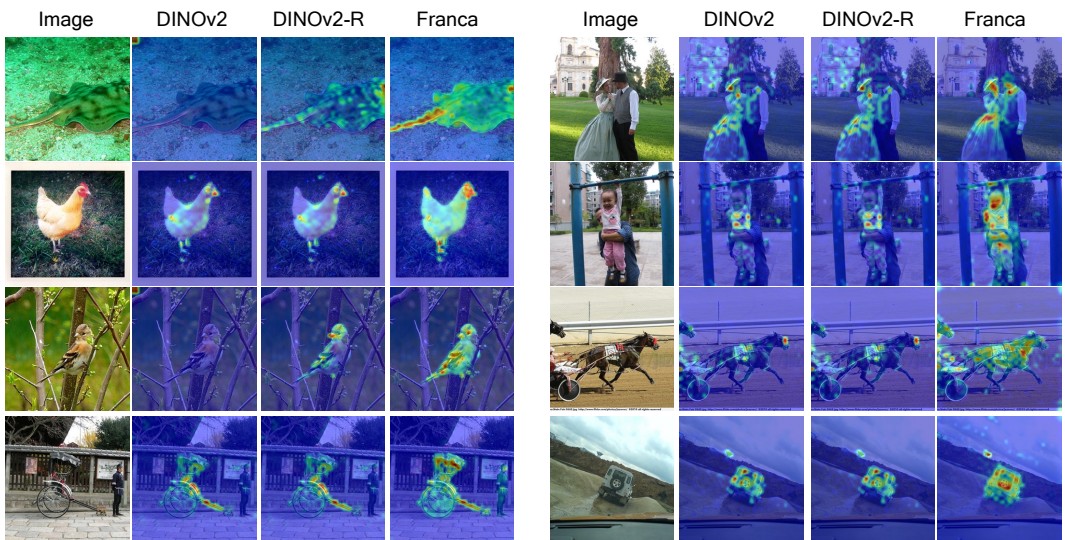

Figure 9: *Self-attention maps* utilizing $14 \times 14$ patches. These maps are visualized using the [CLS] token on the last layer's heads on the validation set of ImageNet-1K (Russakovsky et al., 2015). Franca has better localization than DINOv2 with registers (Darcet et al., 2024) without requiring the use of registers, where the nested Matryoshka clustering captures fine-grained details, e.g., feathers, beaks of bird.

recognition, fine-grained object classification (such as food, cars, and aircraft), and texture recognition. Following (Oquab et al., 2024), we train a logistic regression classifier on features extracted from a frozen backbone. This approach focuses solely on the quality of the visual features and provides a fair way to compare performance across different tasks. Although some of these benchmarks tend to favor models trained with text supervision, our features perform strongly and competitively across many categories.

As shown in Table 6, our method, Franca, transfers well across a wide range of downstream tasks. Our ViT-G/14 model (Franca-G) achieves the same performance as DINOv2-G and outperforms Web-SSL-G by 1.1% despite being trained on much less data. It also matches the performance of larger models like OpenCLIP-G. On datasets such as CIFAR-100 and Oxford Pets, Franca-G achieves 0.7% and 0.5% gains over DINOv2G respectively, demonstrating Franca's strong generalization ability across both natural and fine-grained classification tasks.

## E.2 ATTENTION MAPS VISUALIZATIONS

We compare self-attention maps from the final layer's [CLS] token of DINOv2, DINOv2R (with registers) (Darcet et al., 2024) and Franca in Figure 9. DINOv2 often fails to localize objects, especially under clutter or occlusion, while DINOv2R offers only minor improvements. In con-

| MODEL | ARCH. | DATA | $d=0$ | $d=1$ | $d=2$ | All | SI-RMSE $\downarrow$ |
|---|---|---|---|---|---|---|---|
| IMAGE RESOLUTION $800 \times 800$ (SPAIR-71K) AND $480 \times 480$ (NYUV2) | | | | | | | |
| OpenCLIP | ViT-B/14 | LAION-2B | 18.31 | 16.78 | 17.05 | 17.63 | 0.39 |
| SigLIP2 | ViT-B/14 | WebLI | 9.29 | 5.96 | 5.82 | 7.83 | 0.56 |
| DINOv2$^\dagger$ | ViT-B/14 | IN-21K | 42.82 | 33.38 | 34.82 | 38.71 | 0.33 |
| DINOv2$^\S$ | ViT-B/14 | LVD-142M | 63.04 | **51.09** | **48.76** | **55.98** | **0.25** |
| Franca (ours) | ViT-B/14 | IN-21K | 53.46 | 41.9 | 44.2 | 45.6 | 0.30 |
| OpenCLIP | ViT-L/14 | LAION-2B | 22.73 | 20.18 | 20.37 | 21.26 | 0.37 |
| SigLIP2 | ViT-L/14 | WebLI | 8.66 | 5.38 | 5.05 | 6.88 | 0.55 |
| DINOv2$^\dagger$ | ViT-L/14 | IN-21K | 57.60 | 42.91 | 44.17 | 50.68 | 0.31 |
| DINOv2$^\S$ | ViT-L/14 | LVD-142M | 63.91 | **53.11** | **51.91** | 56.92 | **0.23** |
| Franca (ours) | ViT-L/14 | LAION-600M | 58.50 | 46.74 | 48.28 | 51.76 | 0.25 |

Table 7: *Probing 3D understanding* via keypoint matching on SPair-71K and monocular depth estimation on NYUv2. We report correspondence accuracy (higher is better) under increasing viewpoint changes ($d=0, 1, 2$, and overall) on SPair-71K, and scale-invariant RMSE (lower is better) for depth estimation on NYUv2. Results are shown for high ($800^2$) resolution SPair-71K input. $^\dagger$: reproduced on IN-21K without distillation; $^\S$: distilled from DINOv2-G.

trast, Franca yields sharply focused attention maps aligned with object boundaries, even for small or partially occluded instances. This suggests that our Matryoshka-style multi-head clustering promotes semantically rich features and finer-grained representations.

### E.3 PROBING 3D UNDERSTANDING

We evaluate the geometric understanding of Franca on two tasks: keypoint correspondence (SPair-71k (Min et al., 2019)) and monocular depth estimation (NYUv2 (Silberman et al., 2012)). For SPair-71k, the goal is to establish dense keypoint correspondences under varying viewpoint changes, where we report accuracy across all keypoints and under increasing viewpoint disparity. For depth estimation, we follow the AdaBins protocol (Bhat et al., 2021) and measure performance using the scale-invariant root-mean-square error (SI-RMSE). All evaluations are conducted at high resolution: $800 \times 800$ for SPair-71k and $480 \times 480$ for NYUv2.

As shown in Table 7, Franca achieves strong performance across both tasks. On SPair-71k, it outperforms DINOv2 trained on the ImageNet-21K dataset, both at ViT-B and ViT-L models. On NYUv2, Franca achieves comparable performance as DINOv2 distilled from larger DINOv2 model. This is also due to the fact that DINOv2 uses NYU Depth v2 during pretraining (as part of LVD 142M). These results indicate that Franca learns a spatial representation that captures both fine-grained 2D alignment and coarse 3D structure, generalizing well from object-centric pretraining to downstream geometric tasks.

### E.4 OVERCLUSTERING

We evaluate Franca using the overclustering protocol from (Ziegler & Asano, 2022), which measures semantic alignment of spatial features in a label-free setting. Patch embeddings are clustered with $K$-Means and matched to ground-truth segmentation masks via Hungarian matching (Kuhn, 1955), and performance is reported as mean IoU (mIoU). This task highlights the ability of representations to capture fine-grained structure, which is crucial for dense prediction tasks such as semantic segmentation and object detection.

As shown in Table 8, Franca consistently outperforms strong baselines across backbones and clustering granularities. On ViT-B/14, it achieves the highest VOC performance at $K = 300$ (56.4 mIoU), surpassing DINOv2-B$^\S$ (52.5) while remaining competitive on COCO-Things. With ViT-L/14, Franca delivers its strongest results: it reaches 47.4 and 58.9 mIoU on VOC ($K = 100/K = 300$), and 49.6 and 54.4 on COCO-Things, outperforming DINOv2-L$^\S$, Web-SSL, and SigLIP 2. At ViT-G/14 scale, Franca maintains a clear lead over DINOv2-G and Web-SSL, reaching 49.2 on VOC ($K = 300$) and 31.9 on COCO-Things ($K = 300$), despite the challenging scaling regime.

These results underscore that Franca not only scales effectively to larger backbones but also delivers robust gains on both coarse (VOC) and complex (COCO-Things) datasets. Importantly, it

| METHOD | BACKBONE | OVERCLUSTERING | | | |
|---|---|---|---|---|---|
| | | VOC | | COCO-THINGS | |
| | | $K = 100$ | $K = 300$ | $K = 100$ | $K = 300$ |
| SigLIP | ViT-B/16 | 29.5 | 36.9 | 41.5 | 53.4 |
| iBOT | ViT-B/16 | 21.2 | 29.1 | 18.3 | 26.3 |
| EVA-CLIP | ViT-B/16 | **43.3** | 49.1 | 41.3 | 52.0 |
| DINOv2[†] | ViT-B/14 | 25.9 | 34.7 | 20.5 | 28.7 |
| DINOv2[§] | ViT-B/14 | 39.2 | 52.5 | **46.5** | **54.0** |
| Franca (ours) | ViT-B/14 | 37.5 | **56.4** | 38.8 | 51.1 |
| SigLIP 2 | ViT-L/16 | 24.3 | 40.4 | 43.5 | 50.7 |
| Web-SSL | ViT-L/14 | 28.2 | 37.7 | 26.3 | 33.1 |
| DINOv2[†] | ViT-L/14 | 25.9 | 34.7 | 24.1 | 35.1 |
| DINOv2[§] | ViT-L/14 | 26.5 | 43.0 | 34.8 | 45.7 |
| Franca (ours) | ViT-L/14 | **47.4** | **58.9** | **49.6** | **54.4** |
| Web-SSL | ViT-G/14 | 26.0 | 33.4 | 15.5 | 21.5 |
| DINOv2 | ViT-G/14 | 19.5 | 27.7 | 20.7 | 29.2 |
| Franca (ours) | ViT-G/14 | **39.4** | **49.2** | **25.8** | **31.9** |

Table 8: *OverClustering Performance.* Comparison of overclustering performance (mIoU) on Pascal VOC and COCO-Things datasets with $K = 100$ and $K = 300$. [†]: reproduced on IN-21K, without distillation; § : distilled from DINOv2-G on LVD-142M.

Table 9: *Ablating the Dataset Size used for Training RASA Head*

| | IN-CONTEXT | | LIN. SEG. | | | IN-CONTEXT | | LIN. SEG. | |
|---|---|---|---|---|---|---|---|---|---|
| Fraction | VOC | ADE20K | VOC | ADE20K | Fraction | VOC | ADE20K | VOC | ADE20K |
| 0.1 | 76.6 | 35.2 | 89.3 | 46.2 | 0.1 | **76.7** | 35.3 | **89.4** | **46.1** |
| 0.2 | **76.7** | **35.3** | **89.4** | **46.2** | 0.2 | 76.7 | **35.4** | 89.4 | 45.9 |
| 0.3 | 76.7 | 35.3 | 89.3 | 46.0 | 0.3 | 76.7 | 35.4 | 89.3 | 46.0 |
| 0.4 | 76.7 | 35.3 | 89.3 | 46.1 | 0.4 | 76.7 | 35.4 | 89.4 | 46.0 |
| 0.5 | 76.7 | 35.3 | 89.3 | 46.2 | 0.5 | 76.7 | 35.4 | 89.4 | 45.9 |
| 0.6 | 76.7 | 35.3 | 89.3 | 46.1 | 0.6 | 76.7 | 35.4 | 89.3 | 46.0 |
| 0.8 | 76.7 | 35.3 | 89.3 | 46.2 | 0.8 | 76.7 | 35.4 | 89.4 | 45.9 |
| (a) Fractions of the COCO Dataset | | | | | (b) Fractions of the Imagenet Dataset | | | | |

rivals or surpasses multimodal baselines like EVA-CLIP and SigLIP 2, highlighting its strength as a unimodal self-supervised learner for dense prediction settings.

### E.5 ABLATING RASA

#### E.5.1 DATASET SIZE REQUIREMENTS FOR TRAINING THE RASA HEAD

We ablate the dataset size required for training the **RASA** head by varying the fraction of images sampled from COCO (scene-centric) and IMAGENET100 (object-centric). Tables 9a and 9b summarize the results across in-context segmentation and linear segmentation.

**Observations.** On COCO, as little as $10\%$–$20\%$ of the data suffices to reach peak performance, with VOC and ADE20K scores saturating at 76.6–76.7 and 35.2–35.3 (in-context), and 89.3–89.4 and 46.2 (linear segmentation). Increasing the fraction up to $80\%$ yields no further improvements, indicating that performance plateaus once $\sim$10k images are used.

A similar trend is observed on IMAGENET100, where $10\%$–$20\%$ of the dataset again provides optimal results (76.7 VOC / 35.3–35.4 ADE20K in-context, 89.4 VOC / 45.9–46.1 ADE20K for linear segmentation). Larger fractions show negligible variation, well within noise levels.

**Conclusion.** These findings demonstrate two key points. First, training the RASA head is highly *data-efficient*: roughly 10k images are sufficient to achieve strong disentanglement. Second, the

type of dataset—COCO (scene-oriented) vs. IMAGENET100 (object-oriented)—does not materially affect downstream performance. This indicates that RASA is largely agnostic to the semantic bias of the training distribution and can be reliably trained with minimal data.

Based on these observations, we adopt PASCAL VOC as our default choice for training the RASA head, since it is compact yet sufficient to reach optimal performance. With this setup, we achieve strong results across multiple benchmarks (Table 10). This confirms that a lightweight dataset such as Pascal VOC is adequate for effective training, while maintaining competitive performance across diverse tasks.

Table 10: Training the RASA head on top of VIT-BASE of `Franca` on Pascal VOC.

| SETUP | IN-CONTEXT | | LIN. SEG. | |
|---|---|---|---|---|
| | VOC | ADE20K | VOC | ADE20K |
| `Franca` | 76.2 | 35.0 | 89.4 | 46.0 |
| `Franca` + RASA | **76.7** | **35.3** | **89.4** | **46.0** |

### E.5.2 LEARNING RATE FOR TRAINING THE DUAL-LINEAR POSITION PREDICTOR

We also ablate the learning rate used when training each dual-linear position predictor in the RASA head. Results are reported on both PASCAL VOC and COCO in Tables 11a and 11b respectively.

**Observations.** On PASCAL VOC in Table 11a, the optimal performance is achieved with a learning rate of 0.002, reaching 76.7 VOC / 35.3 ADE20K for in-context and 89.4 VOC / 46.0 ADE20K for linear segmentation. Larger learning rates such as 0.005 yield slightly reduced segmentation accuracy, while very small rates (0.0001) underperform across both tasks. Interestingly, 0.0005 produces a competitive ADE20K score (46.0) but is less stable overall.

On COCO in Table 11b, the trend differs: the best results are obtained at the much smaller learning rate of 0.0001 (76.7 VOC / 35.3 ADE20K in-context, 89.4 VOC / 46.2 ADE20K linear segmentation). Higher learning rates (0.002, 0.005) still maintain strong performance but introduce minor drops or fluctuations, particularly in VOC segmentation. The consistency of 0.0001 across all metrics suggests that COCO benefits from more conservative updates during training.

**Conclusion.** These results indicate that while the RASA head is generally robust to a wide range of learning rates, the optimal configuration is dataset-dependent. For PASCAL VOC, a moderately large learning rate (0.002) works best, while for COCO, a smaller learning rate (0.0001) provides the most stable and accurate results. Overall, this highlights that tuning the learning rate can provide small but measurable gains, though the model remains relatively stable across settings.

### E.5.3 NUMBER OF EPOCHS FOR TRAINING THE DUAL-LINEAR POSITION PREDICTOR

We further ablate the effect of the number of training epochs used for each dual-linear position predictor in the RASA head. Results on both COCO and PASCAL VOC are reported in Tables 12a and 12b.

**Observations.** Across both datasets, performance improves gradually from 1 to 3 epochs, with the most consistent gains observed when training for 5 epochs. On COCO in Table 12a, 5 epochs achieves the strongest overall performance (76.7 VOC / 35.3 ADE20K for in-context, 89.4 VOC / 46.2 ADE20K for linear segmentation). Beyond this point, additional training (7 or 10 epochs) does not yield further benefits and in some cases slightly reduces performance, particularly on ADE20K segmentation.

A similar trend is evident when training on PASCAL VOC in Table 12b. The 5-epoch configuration again provides the best balance across both metrics (76.7 VOC / 35.3 ADE20K in-context, 89.4 VOC / 46.0 ADE20K linear segmentation). Extending to 7 or 10 epochs produces only marginal changes, with scores fluctuating around the same plateau. Interestingly, ADE20K segmentation shows a minor uptick at 10 epochs (46.1), but the difference relative to 5 epochs is negligible and within noise.

Table 11: *Ablating the learning rate used to train each dual linear position predictor.*

| | IN-CONTEXT | | LIN. SEG. | | | IN-CONTEXT | | LIN. SEG. | |
|---|---|---|---|---|---|---|---|---|---|
| lr | VOC | ADE20K | VOC | ADE20K | lr | VOC | ADE20K | VOC | ADE20K |
| 0.002 | **76.7** | **35.3** | **89.4** | **46.0** | 0.002 | 76.7 | 35.3 | 89.4 | 45.9 |
| 0.005 | 76.7 | 35.3 | 89.4 | 45.7 | 0.005 | 76.5 | 35.0 | 89.4 | 46.1 |
| 0.0001 | 76.3 | 34.7 | 89.4 | 45.8 | 0.0001 | **76.7** | **35.3** | **89.4** | **46.2** |
| 0.0005 | 76.6 | 35.2 | 93.9 | 46.0 | 0.0005 | 76.6 | 35.2 | 89.4 | 45.9 |

(a) Learning Rates on the Pascal VOC         (b) Learning Rates on the COCO Dataset

Table 12: *Ablating the number of Epochs used to train each dual linear position predictor.*

| | IN-CONTEXT | | LIN. SEG. | | | IN-CONTEXT | | LIN. SEG. | |
|---|---|---|---|---|---|---|---|---|---|
| Epochs | VOC | ADE20K | VOC | ADE20K | Epochs | VOC | ADE20K | VOC | ADE20K |
| 1 | 76.6 | 35.2 | 89.4 | 46.1 | 1 | 76.1 | 34.5 | 89.4 | 45.9 |
| 2 | 76.6 | 35.3 | 89.4 | 45.8 | 2 | 76.3 | 34.6 | 89.3 | 46.2 |
| 3 | 76.7 | 35.3 | 89.4 | 46.0 | 3 | 76.4 | 34.9 | 89.4 | 45.8 |
| 5 | **76.7** | **35.3** | **89.4** | **46.2** | 5 | **76.7** | **35.3** | **89.4** | 46.0 |
| 7 | 76.6 | 35.2 | 89.4 | 45.9 | 7 | 76.5 | 35.0 | 89.3 | 46.0 |
| 10 | 76.7 | 35.3 | 89.3 | 45.8 | 10 | 76.6 | 35.0 | 89.3 | **46.1** |

(a) Number of Epochs on COCO Dataset         (b) Number of Epochs on Pascal VOC

**Conclusion.** These results highlight that training the dual-linear position predictor is *computationally efficient*, requiring only around 5 epochs to converge. Extending beyond 5 epochs yields no meaningful improvements and may even slightly degrade results. Moreover, the stability of the trends across both COCO (scene-centric) and PASCAL VOC (object-centric) datasets reinforces the robustness of this setting. In practice, we adopt 5 epochs as the default configuration for RASA training.

### E.5.4 NUMBER OF ITERATIONS FOR TRAINING THE RASA HEAD

Finally, we ablate the number of *iterations* used to train the RASA head, where each iteration corresponds to training an additional dual-linear position predictor and integrating the previously trained predictors into the final ViT block. Results on PASCAL VOC and COCO are reported in Tables 13a and 13b.

**Observations.** Even a small number of iterations (1–2) produces competitive results across both datasets, suggesting that the RASA head can already provide strong positional disentanglement with minimal overhead. For example, with only 2 iterations we obtain 76.2 VOC / 34.5 ADE20K (in-context) and 89.4 VOC / 46.1 ADE20K (linear segmentation) on PASCAL VOC, which is close to the final performance.

However, performance steadily improves as the number of iterations increases, with the most notable gains occurring between 2 and 8 iterations. At 8 iterations, results peak at 76.7 VOC / 35.3 ADE20K in-context and 89.4 VOC / 46.0 ADE20K for segmentation on PASCAL VOC (in Table 13a), and 76.7 VOC / 35.3 ADE20K in-context and 89.4 VOC / 46.2 ADE20K on COCO (in Table 13b). Beyond this point (10 iterations), performance saturates and in some cases slightly declines, suggesting diminishing returns from further iterative training.

Interestingly, across both datasets, the segmentation metrics appear to plateau earlier (around 4–6 iterations), while in-context segmentation continues to benefit until iteration 8. This indicates that additional iterations primarily refine segmentation-related representations, even though these could be sort of learned by the linear layer used in linear segmentation.

**Conclusion.** These experiments demonstrate that the RASA head achieves strong results even with a small number of iterations, but the optimal trade-off between performance and compute is obtained

Table 13: *Ablating the number of dual predictor layers used for training RASA head (i.e., the number of iterations).*

| Iters | IN-CONTEXT | | LIN. SEG. | | Iters | IN-CONTEXT | | LIN. SEG. | |
|---|---|---|---|---|---|---|---|---|---|
| | VOC | ADE20K | VOC | ADE20K | | VOC | ADE20K | VOC | ADE20K |
| 1 | 76.2 | 34.2 | 89.3 | 46.1 | 1 | 76.1 | 34.2 | 89.3 | 45.9 |
| 2 | 76.2 | 34.5 | 89.4 | **46.1** | 2 | 76.2 | 34.5 | 89.4 | 46.1 |
| 3 | 76.4 | 34.5 | 89.3 | 45.9 | 3 | 76.3 | 34.5 | 89.4 | 46.0 |
| 4 | 76.5 | 34.6 | 89.3 | 45.9 | 4 | 76.5 | 34.5 | 89.4 | 46.2 |
| 5 | 76.3 | 34.5 | 89.3 | 46.0 | 5 | 76.3 | 34.5 | 89.4 | 46.0 |
| 6 | 76.4 | 34.8 | 89.4 | 45.8 | 6 | 76.4 | 34.8 | 89.4 | 46.0 |
| 8 | **76.7** | **35.3** | **89.4** | 46.0 | 8 | **76.7** | **35.3** | **89.4** | **46.2** |
| 10 | 76.3 | 35.1 | 89.3 | 46.0 | 10 | 76.6 | 35.2 | 89.4 | 46.0 |

| (a) Number of Iterations with Pascal VOC | (b) Number of Iterations with Coco |
|---|---|

at 8 iterations. Beyond this, additional predictors provide no meaningful improvements, confirming that the gains saturate after moderate iterative refinement.

E.6 COMPUTATIONAL COST

We report the computational cost of both pretraining and incremental high-resolution finetuning in terms of GPU-hours. For ViT-B, ViT-L, and ViT-G, we provide estimates covering the entire training pipeline, including the staged finetuning at $364 \times 364$ and $518 \times 518$ resolutions. The total GPU-hours for each model are summarized in Table 14.

A few observations can be drawn from these numbers. First, the compute cost scales linearly with model size: Franca-G requires roughly $\sim 13\times$ the GPU-hours of Franca-B, while Franca-L lies in between with $\sim 4\times$ Franca-B. This reflects both the higher per-step cost of larger models and the greater degree of parallelism (more GPUs per run) needed to sustain efficient training throughput. Second, compared to prior large-scale self-supervised training efforts such as Meta-CLIP and DINOv2-G, our Franca models are trained with significantly lower total GPU budgets, highlighting the efficiency benefits of our setup. For example, Franca-G was trained with only $\sim 20$K GPU-hours on H100s, which is an order of magnitude smaller than the 368K GPU-hours reported for MetaCLIP despite comparable training horizons.

Importantly, the reported numbers include practical overheads such as hyper-parameter searches, checkpoint restarts, and failed runs. This makes our accounting closer to the actual wall-clock compute footprint of the project, rather than an idealized lower bound. We note that high-resolution finetuning contributes a non-negligible fraction of the total cost, especially for Franca-L and Franca-G, underscoring the importance of carefully balancing resolution scaling against available compute resources.

| Model Arch. | GPU type | # GPUs | Iterations (Pretrain+ Finetune) | GPU hours |
|---|---|---|---|---|
| MetaCLIP | A100 | 256 | 390K | 368,640 |
| DINOv2-G | A100 | 128 | 625K + 20K | 22,016 |
| Franca-B | H100 | 32 | 625K + 40K | 1,504 |
| Franca-L | H100 | 64 | 625K + 40K | 5,952 |
| Franca-G | H100 | 128 | 625K + 40K | 19,992 |

Table 14: Compute cost in GPU-hours for pretraining and high-resolution finetuning. We estimate the overall GPU hours for our project, including hyper-parameter search, restarts from checkpoints, and failed runs, amounting to a total of 160K GPU-hours on NVIDIA H100 GPUs (80GB, NVLink interconnect). The table provides a breakdown across model scales (ViT-B, ViT-L, ViT-G) and training phases (pretraining vs. finetuning). Reported values reflect actual wall-clock consumption rather than idealized FLOP counts.

