# OpenReview forum: "Franca:  Nested Matryoshka Clustering for Scalable Visual Representation Learning"
_ICLR.cc/2026/Conference — ICLR 2026 Conference Withdrawn Submission_

### Official Review · Reviewer_Wzrs · 2025-10-15

**Soundness:** 2
**Presentation:** 2
**Contribution:** 2
**Rating:** 2
**Confidence:** 4

**Summary:**

The paper introduces Franca, the first fully open-source vision foundation model that matches or exceeds the performance of proprietary models like DINOv2, CLIP, and SigLIP2 using public datasets (Imagenet-21K and LAION-600M). It proposes innovative techniques including Matryoshka multi-head clustering for efficient multi-granular representations, CyclicMask to balance spatial patch distribution, and RASA to remove positional biases, demonstrating consistent performance gains across tasks like in-context learning and segmentation.

**Strengths:**

The paper’s key strength lies in its commitment to full openness (data, code, weights), enabling reproducibility and community access, which is a significant advancement in vision foundation model research.

**Weaknesses:**

1. The paper adopts the Matryoshka concept (Kusupati et al., 2022) to create nested, multi-granular representations, which is a practical extension rather than a novel theoretical framework. The idea of slicing feature dimensions (e.g., $d, d/2, d/4, \ldots$) and applying independent projection heads is described mechanically, but there’s little discussion on why this hierarchical structure inherently improves learning or how it addresses the theoretical limitations of large codebooks (e.g., Sinkhorn-Knopp’s ambiguity).

2. The paper adopts the Matryoshka concept (Kusupati et al., 2022) to create nested, multi-granular representations, which is a practical extension rather than a novel theoretical framework. The idea of slicing feature dimensions (e.g., $d, d/2, d/4, \ldots$) and applying independent projection heads is described mechanically, but there’s little discussion on why this hierarchical structure inherently improves learning or how it addresses the theoretical limitations of large codebooks (e.g., Sinkhorn-Knopp’s ambiguity).

3. RASA aims to disentangle positional and semantic information through a linear projection and subtraction process (Equations 1-4, Figure 6). The method is mathematically defined, using Gram-Schmidt orthonormalization and iterative refinement, but the theoretical basis for its effectiveness is underdeveloped. For instance, there’s no analysis of why linear predictability of patch coordinates is a sufficient proxy for all positional biases, or how this affects the geometry of the learned feature space (beyond the entropy metric in Figure 5).

4. The paper builds on DINOv2 (Oquab et al., 2024) and extends it with these components, but the theoretical contribution is framed as an incremental improvement rather than a paradigm shift. The abstract and introduction highlight practical goals (open-source, scalability), but do not delve into fundamental questions about self-supervised learning (SSL), such as the role of clustering ambiguity or the interplay between spatial and semantic features.

**Questions:**

1. The qualitative results (e.g., PCA slices in Figure 3) show Franca preserving semantic structure better than DINOv2 at lower dimensions (e.g., dim/16), which is promising. Similarly, the entropy analysis (Figure 5) indicates that RASA post-training increases spatial entropy, suggesting reduced positional bias. However, the quantitative results in Table 2 and Figure 2b show that Franca’s improvements over DINOv2 are incremental rather than dominant. For instance, in in-context segmentation (mIoU) on Pascal VOC, Franca (ViT-L/14) achieves 79.5% compared to DINOv2’s 74.6% (a 4.9% gain), but on other tasks like linear segmentation (VOC), the gains are smaller (89.4% vs. 90.3% for DINOv2). This suggests that while qualitative improvements are notable, they don’t translate into uniformly superior quantitative performance across all metrics. Why so?

2. The paper excels in empirical validation, with controlled comparisons (Figure 2b), ablation studies (Figure 2a), and diverse benchmarks (Table 2). The incremental gains (e.g., 12.8% total improvement in in-context segmentation) and qualitative visualizations (Figures 3, 5) provide strong evidence of utility. However, this empirical strength contrasts with the lack of theoretical depth. For example, while Figure 5 shows RASA increasing spatial entropy, there’s no model or hypothesis explaining why this entropy correlates with better semantic representation, beyond the intuitive notion of reducing bias.

---

### Official Review · Reviewer_cPWT · 2025-10-27

**Soundness:** 2
**Presentation:** 2
**Contribution:** 2
**Rating:** 4
**Confidence:** 4

**Summary:**

This paper presents a new open-source vision foundation model, named Franca. The method introduces two main technical components: 1) a Matryoshka multi-head clustering approach intended to refine features into fine-grained clusters without increasing model size, and 2) a post-pretraining strategy, RASA, designed to remove spatial biases from representations. The authors evaluate the effectiveness and competitiveness of the proposed method through experiments across diverse tasks, including classification, dense prediction tasks, out-of-distribution detection, and 3D understanding.

**Strengths:**

1. The nested Matryoshka clustering projector enables coarse-to-fine granular representations without increasing model size, which provides a significant performance gain on in-context learning tasks (e.g., increasing In-Context mIoU from 69.8 to 73.7).
2. The model's effectiveness is validated through comprehensive experiments across a diverse range of benchmarks, including dense prediction tasks, out-of-distribution detection, 3D understanding, classification and robustness.
3. The paper provides a clear ablation study showing the incremental contributions of its main components (CyclicMask, Matryoshka, RASA) and includes a detailed analysis of RASA's hyperparameters, such as dataset size, learning rate, training epochs, and iterations.

**Weaknesses:**

1. There is a notable discrepancy between the paper's textual claims regarding robustness and the corresponding experimental data. Specifically, Section 3.3 claims that "Franca-G matches DINOv2-G across all three datasets". This assertion is not fully supported by the results in Table 3. The data shows that DINOv2-G achieves clearly higher scores on two of these benchmarks: ImageNet-R (78.8 vs. 75.8) and Sketch (62.5 vs. 60.6). Such overstatements weaken the credibility of the paper's positioning against state-of-the-art models in this specific domain.
2. The paper positions RASA as one of its two key technical innovations. However, its empirical contribution, as demonstrated in the main ablation study (Figure 2a), is marginal. RASA accounts for only a +0.3 mIoU gain (from 76.2 to 76.5), which is negligible compared to the +2.5 mIoU gain from the standard, non-novel "High-Res FT" step. A core technical contribution that fails to demonstrate a meaningful impact in the paper's own ablations severely weakens the paper's overall claim to novelty.
3. The paper introduces CyclicMask as a component of its method, yet the ablation study in Figure 2a fails to provide a compelling justification for its inclusion. The component yields zero gain for in-context segmentation (remaining at 69.8 mIoU) and a trivial 0.2% gain for linear probing. Adding a technical component that offers no significant empirical benefit introduces unnecessary complexity, suggesting an effort to artificially bolster the paper's perceived methodological novelty rather than delivering tangible improvements.
4. The authors acknowledge DINOv3 as "concurrent development"  but stop at preliminary observations, lacking any substantial experimental comparison or in-depth analysis. Given DINOv3 is a directly relevant and state-of-the-art work, this omission makes it difficult to assess Franca's claimed advancements and positioning convincingly.
5. The Method section (Section 2) lacks clarity and fails to provide a clear explanation of the proposed framework. The Preliminaries part inadequately introduces foundational concepts by assuming expert knowledge of specific prior works like DINO-style and iBOT-style heads. The paper frequently delves into technical details without adequate motivational context. These make it difficult to follow the logical flow of the method. Regarding the main overview figure (Figure 1), it dedicates two-thirds of its space to data pipelines and a results plot rather than focusing on the technical architecture; and the left panel only illustrates a schematic of one core component, which is insufficient for a comprehensive understanding.
6. In Table 3, the footnote † is used to denote models "reproduced on IN-21K without distillation", which is applied to DINOv2. The "Franca (ours)" ViT-B/14 model was also trained on IN-21K without distillation. To ensure a clear and fair like-for-like comparison, should the † mark also be applied to the Franca (ours) ViT-B/14 row?

**Questions:**

See Weaknesses.

---

### Official Review · Reviewer_jGMM · 2025-10-28

**Soundness:** 2
**Presentation:** 2
**Contribution:** 1
**Rating:** 2
**Confidence:** 3

**Summary:**

This paper introduces Franca, a self-supervised vision foundation model that aims to match or surpass the performance of state-of-the-art proprietary models like DINOv2. A key contribution emphasized by the authors is that Franca is "fully open-source," with claims of releasing training code, data details, and model weights. On the technical side, the paper introduces two innovations: (1) a Matryoshka multi-head clustering projector to replace standard Sinkhorn-Knopp-based clustering, and (2) a post-pretraining technique called RASA (Removal of Absolute Spatial Attributes) to reduce positional bias in the learned representations. The resulting model, Franca, is evaluated on a wide range of downstream tasks, including segmentation, object discovery, classification, robustness, and 3D awareness, where it demonstrates strong performance.

**Strengths:**

- The ambition to create a high-performing, fully open-source vision foundation model is a valuable goal that promotes reproducibility and accessibility.
- The technical motivations are sound, addressing limitations of prior works such as semantic ambiguity in clustering (via Matryoshka heads) and positional bias in ViT features (via RASA).
- The model is evaluated on a comprehensive suite of benchmarks beyond classification, including dense segmentation, video, robustness, and 3D awareness, providing a holistic view of its capabilities.

**Weaknesses:**

- **Insufficient Analysis and Ablation:** My primary concern is that the paper lacks the deep analysis on its design choice, which is expected for a research publication; as a consequence, the paper reads more like a technical report. See also questions.
    - **Matryoshka Clustering:** The paper claims the multi-head clustering is "parameter-efficient" but provides no explanation or quantification of these efficiency gains compared to the Sinkhorn-Knopp clustering it replaces. Furthermore, there are no ablation studies on its design, such as the choice of nested dimensions ($\mathcal{M}$), making it difficult to assess the contribution.
    - **RASA:** The performance gain from RASA appears marginal in the paper's own ablation study (Figure 2a). While Figure 5b shows RASA increases spatial entropy, the paper does not sufficiently connect this intermediate result to a significant impact on downstream tasks.
    - **Masking:** Figure 4 visualizes several masking strategies, but only CyclicMask is empirically compared to the baseline (Figure 2a), where it shows a very small gain. The other strategies are not evaluated.
- **Overstated and Unverifiable Contributions:**
    - **Scientific Contribution:** The "open-source" nature of the model is repeatedly emphasized as a primary contribution. While commendable, releasing code and weights for a reproduced/improved model is a valuable community contribution but is not, by itself, a significant *scientific* contribution. The paper must be judged on the novelty and insights of its techniques.
    - **Verifiability:** All claims of transparency are currently unverifiable. The submission (in OpenReview) was not accompanied by code, weights, or anonymized repository. Also, claims about releasing intermediate checkpoints to provide "unique insight" are unsubstantiated, as no such insights are actually presented in the paper.
- **Inconsistent Claims and Results:**
    - The paper claims "performance gains of Franca increase with model capacity". However, the provided results in Table 2 do not consistently support this. In Table 2a, a direct comparison is not possible, as the ViT-G model skipped the high-resolution fine-tuning stage that the ViT-B/L models received. Therefore, there are currently only two datapoints (ViT-B and L), which is too weak to support the observed trend.
    - More importantly, Table 2b (Video Object Segmentation) shows a direct contradiction: the ViT-B/14 model (70.6) outperforms the larger ViT-L/14 model (69.1).

**Questions:**

- **Efficiency:** Could the authors please quantify the efficiency gains (in parameters and/or computational cost) of the proposed Matryoshka clustering projector compared to the optimal-transport-based clustering used in DINOv2?
- **Generalization Claim:** The authors hypothesize that DINOv2's use of large codebooks "may not generalize well across domains" (L80). Is there empirical evidence to support this claim? If so, do the experiments demonstrate that Franca's multi-granular clustering specifically resolves this cross-domain generalization problem?
- **Scaling Claim:** Given the contradictory result in Table 2b (where ViT-B > ViT-L for VOS), could the authors strengthen the claim about performance scaling with model capacity (L352)? A controlled comparison (e.g., ViT-B, L, and G all *without* high-resolution adaptation) would be needed.
- **Intermediate Checkpoints:** The paper states that the release of intermediate checkpoints is a key aspect of the contribution (L53). What specific, concrete scientific insights (e.g., regarding convergence, emergent properties, etc.) would be derived from analyzing these checkpoints? If so, it would be beneficial to include them in the paper.
- **Feature Coherence:** In Figure 7 , the authors claim that Franca produces more coherent color segments than DINOv2. What is the primary component responsible for this qualitative improvement, and what quantitative metric does this observation correlate with?
- **Benchmark Justification:** Could the authors briefly justify the choice of the Hummingbird and Feat2GS benchmarks? What specific properties do they evaluate that are not captured by more traditional segmentation or 3D benchmarks?
- **Distillation from ViT-G**: The paper claims that DINOv2 largely benefits from distillation, which Franca does not employ (L361, L414, L421). Would Franca also benefit from distillation?

Minor suggestions

- **Baseline Description:** The model "DINOv2-R" (presumably DINOv2 with register) is used in Table 2b, but it is not defined or described before.
- **Typo**: The citation should be (Darcet et al., 2025) instead of (Darcet et al., 2024) in L207.

---

### Official Review · Reviewer_4qbD · 2025-10-31

**Soundness:** 3
**Presentation:** 3
**Contribution:** 3
**Rating:** 6
**Confidence:** 4

**Summary:**

The paper presents Franca, a self-supervised model trained on  minimally curated data that shows improvements over baselines such as DINOv2 and CLIP. Franca is built upon DINOv2 and adds additional components to improve the learning process. These components include Matryoska representations from Kusupati et al. to improve the clustering in DINOv2, CyclicMask from Darcet et al., 2024 to improve patch distribution and RASA, a new post-training approach to remove positional bias in the features.

The paper provides experimental results to demonstrate Franca's effectiveness compared to existing methods.

Different from existing works, Franca open sources all its data, code and models, facilitating research towards more reproducible foundation models.

**Strengths:**

The additional components that Franca introduces upon DINOv2 are sound and reasonable. Hierarchical clustering with Matryoska representations is shown to bring significant improvements in Figure 2a. Using CyclicMask is both sound and slightly effective as shown in Figure 2a. RASA is also shown to be effective and produces cleaner features in Figure 7.

The paper provides extensive experimental results in different benchmarks, using different model sizes to compare Franca to baselines. The results show Franca achieving good results compared to the baselines.

Different from other works, Franca open sources all its data, code and models, facilitating research towards more reproducible foundation models.

**Weaknesses:**

The proposed RASA method sounds interesting but appears a bit ad-hoc. I understand that RASA is trying to solve a nested optimization problem on 2 parameters W and Z with the former trying to minimize the L_pos loss while the latter attemps to maximize the loss. However, the update rule only returns an `acceptable` solution where \sigma(WZ) is 1/2. What worse is that amongst the `acceptable` solution is Z=0. It is not clear how to make sure that Z does not converge to 0 during these updates. The paper states that L_pos `saturates` after 9 updates but I am a bit skeptical of this.

In order to demonstrate effectiveness of a foundation model, it is necessary to evaluate them on a variety of benchmarks. Franca provides entensive results on non-parametric benchmarks such as knn classification, in-context segmentation, video object segmentation, and unsupervised object discovery as well as some linear probing on ImageNet and semantic segmentation. However, there is a wide range of benchmarks that were considered in DINOv2 but are missing from Franca's evaluation such as fine-grained classification, long-tailed classification (iNaturalist), video classification, image retrieval and depth estimation.

**Questions:**

The paper claims in L.313 that for ViT-B and ViT-L comparison, the paper uses the same set of hyper-parameters for both Franca and DINOv2. Are these parameters optimized for Franca or DINOv2?

RASA is applied post-training on PASCAL VOC dataset. Why not applying it on the same pre-training dataset? Why this particular choice of dataset? Does dataset choice affect the performance of RASA alot?

There are a few differences between DINOv2's performance reported in the DINOv2 paper and the current paper. For examples, DINOv2 reported 83.5, 86.5, 78.4 with ViT-g for knn, val and v2 benchmarks but Franca reports 83.1, 86.0, 77.9 in Table 3 which makes Franca appears a little bit better.  Similarly, DINOv2 reported 49.0 and 83,0 mIoU for ViT-g resp. on ADE20K and VOC but Franca reports 46.2 and 90.6. What is the reason behind these discrepancies?

For 3d awareness probing, the paper chooses to evaluate on novel view synthesis which is arguably less direct than depth estimation. What is the reason for this choice? I think it is good to include evaluation on depth estimation.

The paper claims that Franca supports hierarchical learning. It would be nice to have at least a qualitative evaluation showing this effect in practice.

---

### Note · Authors · 2025-11-14

I have read and agree with the venue's withdrawal policy on behalf of myself and my co-authors.